# Evolution of oxygen, stratification and their relationship in the North Pacific Ocean in CMIP6 Earth System Models

Lyuba Novi[1], Annalisa Bracco[1], Takamitsu Ito[1], Yohei Takano[2,3]

[1]School of Earth and Atmospheric Sciences, Georgia Institute of Technology, Atlanta, GA, USA.
[2]British Antarctic Survey, Cambridge, UK.
[3]Los Alamos National Laboratory, Los Alamos, NM, USA.

*Correspondence to*: L. Novi (lnovi3@gatech.edu)

**Abstract**. This study examines the linkages between the upper ocean (0-200 m) oxygen ($O_2$) content and stratification in the North Pacific Ocean in four Earth system models (ESMs), an ocean hindcast simulation, and an ocean reanalysis. Trend and variability of oceanic $O_2$ content are driven by the imbalance between physical supply and biological demand. The physical supply is primarily controlled by ocean ventilation, which is responsible for the transport of $O_2$-rich surface waters into the subsurface. Isopycnic Potential Vorticity (IPV), a quasi-conservative tracer proportional to density stratification that can be evaluated from temperature and salinity measurements, is here used as a dynamical proxy for ocean ventilation. The predictability potential of the IPV field is evaluated through its information entropy. Results highlight a strong $O_2$-IPV connection and somewhat higher (than in rest of the basin) predictability potential for IPV across the tropical Pacific, where the El Niño Southern Oscillation occurs. This pattern of higher predictability and strong anticorrelation between $O_2$ and stratification is robust across multiple models and datasets. In contrast, IPV at mid-latitudes has low predictability potential and its center of action differs from that of $O_2$. In addition, the locations of extreme events or hotspots may or may not differ among the two fields, with a strong model dependency, which persists in future projections. These results, on one hand, suggest that in the tropical Pacific may be possible to monitor ocean $O_2$ through few observational sites co-located with the more abundant IPV measurements, and, on the other, question the robustness of the IPV-$O_2$ relationship in the extra-tropics. The proposed framework helps characterizing and interpreting $O_2$ variability in relation to physical variability, and may be especially useful in the analysis of new observationally-based data products derived from the BGC-ARGO float array in combination with the traditional but far more abundant ARGO data.

## 1 Introduction

Dissolved oxygen ($O_2$) in the oceans is crucial for biogeochemical cycling, marine ecosystem and redox chemistry of seawater. $O_2$ is a key element for the survival and functioning of marine organisms as fish, shellfish, marine mammals, and other aquatic life rely on $O_2$ to breathe and carry out essential metabolic processes. Growth, reproduction, and overall health of marine organisms depends on the balance between metabolic demands and $O_2$ supply (Deutsch et al., 2015).

Ocean deoxygenation refers to the long-term decrease in the concentration of $O_2$ in the Earth's oceans. At the global scale, the $O_2$ inventory has been declining significantly over the past decades according to historical observations (Ito et al., 2017; Schmidtko et al., 2017). Changes in $O_2$ concentrations can reflect the impacts of climate change, nutrient pollution, eutrophication, and other human-induced stressors (e.g. Breitburg et al., 2018). Predicting $O_2$ levels in the oceans is especially important around and within Oxygen Minimum Zones (OMZs), which are characterized by layers in the water column with very low $O_2$ concentration due to biological, chemical, and physical processes. As oceans warm, OMZs are posed to increase in number and size across the globe, threatening marine ecosystems. In the North Pacific, a large OMZ exists on the eastern side of the tropical Pacific, and its variability and trends are important also for nitrogen cycling and production of $N_2O$, a potent greenhouse gas (Nevison et al., 2003; Yang et al., 2017). Oxygen measurements, however, are sparse in time and space, and trends remain uncertain, with the decline between 1970 and 2010 estimated to be around $-0.48 \pm 0.35$ % per decade in the upper 1,000m (Bindoff et al., 2019). The uncertainty in the ocean deoxygenation estimates is due to different data sources, interpolation methods, data quality control standards, and data sources, the latter varying from shipboard measurements (ship-based bottle measurements and CTD-$O_2$ profiles) and biogeochemical Argo floats (Roemmich et al., 2019).

Interpreting changes in $O_2$ concentrations requires understandings how ocean circulation, mixing, air-sea gas exchange, biological productivity and respiration operate. The air-sea gas exchange for $O_2$ is relatively efficient, and it maintains the surface water close to saturation with the overlying atmosphere for ice-free regions. Ocean circulation is the primary pathway through which $O_2$ is supplied (or ventilated) into the thermocline and deep ocean. In the subsurface, $O_2$ is gradually consumed by respiration due to the decomposition of dissolved and particulate organic matter. The $O_2$ concentration progressively decreases as water masses age. At climatological timescale, the rates of $O_2$ supply and consumption are balanced to sustain a steady state. In another words, changes in $O_2$ concentration are caused by an imbalance between $O_2$ supply and $O_2$ consumption. On the supply side, the ventilation of $O_2$ is essentially controlled by the ocean circulation and mixing processes. Broadly, ventilation refers to the exchange of waters between the surface layer and the ocean interior (Talley et al., 2011), and involves a wide range of physical processes such as the wind-driven shallow overturning associated with the Subtropical cells (Brandt et al., 2015; Duteil et al., 2014; Eddebbar et al., 2019), the formation of mode and intermediate waters (Claret et al., 2018; Sallee et al., 2010, 2012; Gnanadesikan et al., 2012) and the lateral (isopycnal) eddy stirring (Rudnickas et al., 2019; Gnanadesikan et al., 2013, 2015). These circulation systems are ultimately driven by the atmospheric winds and air-sea buoyancy fluxes which exhibit significant interannual, decadal and multi-decadal variability. Fluctuations in ventilation rates as well as ocean stratification are known to impact both $O_2$ levels (Ridder & England, 2014; Duteil et al., 2014; McKinley et al., 2003) and the distribution of isopycnal potential vorticity (IPV), a dynamical tracer which is proportional to the local stratification and the Coriolis parameter. The use of the absolute value of the Coriolis parameter in the formula, indicated by *, guarantees that the relationship with stratification holds with the same sign in both hemispheres, so that higher IPV* indicates stronger stratification and vice versa. A strong winter-time convective mixing

will produce weakly stratified, $O_2$-rich water masses (low IPV* and high $O_2$), and vice versa. These properties are then brought together into the ocean interior following the pathway of large-scale ocean currents.

In this study, we build upon this relationship and explore the overarching hypothesis that isopycnic potential vorticity (IPV*) may be used as a proxy for $O_2$ with a focus on the North Pacific basin. If this was the case, then IPV* may provide a path to monitor and predict the evolution of $O_2$. In the North Pacific, the Pacific Decadal Oscillation (PDO) is the mode of climate variability that exerts the greatest control on stratification and $O_2$, as shown in Ito et al. (2019). Indeed, the dominant mode of oxygen variability in the North Pacific Ocean is correlated with the PDO index such that the PDO explains about 25% of its variance. In the tropics, the PDO modulates the depths of isopycnal surfaces and biological productivity/respiration together with the El Niño Southern Oscillation (ENSO), while at mid-latitudes is the dominant mode influencing the depth of the winter mixed-layer ventilation and the ventilation processes. Here we analyze outputs from the Coupled Model Intercomparison Project Phase 6 (CMIP6, Eyring, 2016), a major international effort with the primary objective of providing a standardized framework for simulating past, present, and future climate conditions. The participating modeling groups run their climate models under prescribed forcing fields and following a common protocol, and generate a comprehensive set of output datasets freely available to the scientific community through data portals and archives provided by the Earth System Grid Federation (ESGF). Using a suite of Earth System Models (ESMs) run with prescribed carbon dioxide concentrations, we address the following questions:

- How robust is the relationship between $O_2$ and IPV* in the North Pacific across several ESMs and how may it evolve by the end of the 21$^{st}$ century? ($\rightarrow$ HYP 1)

- What are the linkages between $O_2$ and IPV* versus large-scale modes of climate variability such as PDO and ENSO? ($\rightarrow$ HYP 2)

- Where are the hotspots of changes in IPV* and $O_2$, both in the historical period and in the projections, and are they co-located or differ in space and time? ($\rightarrow$ HYP 3)

Our specific objectives are to evaluate the hypotheses that (HYP 1) the ocean ventilation (IPV*) regulates $O_2$ variability in the North Pacific; (HYP 2) the PDO/ENSO-ventilation-$O_2$ linkage provides the basis for the predictability of $O_2$ whenever IPV* is predictable; and (HYP 3) the linkage can be exploited to identify hotspots of $O_2$ changes in variability, means and extremes (see Methods). While testing these hypotheses, we also aim at introducing recently developed approaches for model intercomparison and more generally data analysis to the ocean biogeochemistry community. Specifically, in this work we adopt the information entropy (IE, Prado et al., 2020) concept for evaluating predictability, a data-mining tool for dimensionality reduction and network analysis (δ-MAPS, Fountalis et al., 2018), and the standard Euclidean distance index

(SED, Diffenbaugh and Giorgi, 2012) to evaluate changes in the fields we analyze. In short, Information Entropy (IE) is a metric that measures the amount of randomness and therefore unpredictability in a dataset. For example, if a time series is a random sequence, its entropy will be high, while if a time series follows a sinusoidal curve, the IE will be low. δ-MAPS, on the other hand, combines feature extraction and network analysis into a single framework. Its goal is to identify key features and to visualize how those features relate to one another. Finally, the Standard Euclidean Distance (SED) index is a simple and flexible method used to detect *total* changes in one or more variables in a given dataset (in other words to identify regions that stand out for changes in means, extremes and variability), through measuring the distance in multi-variate space between a baseline period and any other.

After introducing these tools in more detail in section 2, a description of the data analyzed in this work follows in section 3. Results are then presented in section 4, with a general discussion and conclusions to close.

## 2 Materials and methods

In this section, we describe the three tools recently developed for climate science applications and adopted in our analysis and how we calculate IPV*. The Information Entropy (IE) is used to evaluate predictability. IE is defined following Prado et al. (2020) and is based on the recurrence of microstates in a recurrence plot (RP). A RP (Eckmann et al., 1987) is a visualization technique for trajectory recurrence of a given dynamical system described in phase space by a matrix $R_{ij}$ such that

$$R_{ij}(\epsilon) = \Theta\left(\epsilon - |x_i - x_j|\right), \; x_i \in \mathbb{R}^d, \; i,j = 1,2,\dots, K, \tag{1}$$

where $\Theta$ is the Heaviside function, $|\;\;|$ is an Euclidean distance, in our work $x_i$ and $x_j$ are states at time steps i and j, $\varepsilon$ is a threshold distance (the maximum distance between two states to be considered mutually recurrent), $d$ is the $x_i$ space dimension and K is the number of states considered (the length of each analyzed time series). $R_{ij}$ is a matrix which represents non-recurrent (as zeros) and recurrent (as ones) states in phase space respectively, and it is explicitly dependent on ε. Corso et al. (2018) introduced the *Recurrence Entropy* quantifier, for which for a given time series, the probability of occurrence of microstates in its RP is quantified without the need for a space-state reconstruction. A microstate of dimension N is a NxN matrix sampled inside the RP, with probability of occurrence $P_k = n_k/N_{tot}$, where $n_k$ are the number of occurrences of the

microstate, and $N_{tot}$ is the total number of possible configurations of 0 and 1 of the microstate (see Ikuyajolu et al (2021) and Prado et al (2020) for more details). The information entropy IE is then defined as

$$IE(N_{tot}) = - \sum_{k=1}^{N_{tot}} P_k \ln P_k \qquad (2)$$

where k refers to the $k^{th}$ microstate. When IE is normalized by the maximum entropy (corresponding to when all microstates show the same probability) then IE=0 corresponds to perfect predictability, while IE=1 represents chaos. Furthermore, the explicit dependence of the entropy quantifier on ε is removed using the maximum entropy formulation. Prado et al. (2020) have shown that a value for which IE is maximum exists, does not vary much for varying ε and is strongly correlated with the Lyapunov coefficient of the system. We refer the reader to Ikuyajolu et al (2021) for the details of the heuristic used to

estimate the maximum entropy. In essence, Prado et al. (2020) suggest a technique to eliminate the dependence of the entropy computation on the selection of a distance threshold (ε) by finding a clearly defined maximum ($S_{max}$) in the relationship between ε and the entropy (see Fig. 4 in Prado et al. (2020)). This maximum is robust and relatively stable in a range of ε values. Furthermore, there is a strong correlation between the maximum entropy and the Lyapunov exponent. In our work, the code used to compute the entropy (see Data Availability section) uses the heuristic explained in Ikuyajolu et al.

(2021) for the calculation of $S_{max}$ through an iterative procedure that calculates the recurrence entropy for varying ε until a maximum is found and retained. This algorithm requires three input parameters: the microstate dimension (set at 4 in this work, but we explored other values as shown in Results), the number of random samples to compute the microstates distribution in the RP (here 10000) and a random sub-sample used to determine the ε for which entropy is maximum (here 1000). We compute the entropy field of the deseasonalized and detrended IPV* (full signal) using monthly data over the

whole historical and future periods. In each point, the entropy of the IPV* field is associated with recurring microstates in its time series (that define the system and thereby impacts its predictability). The higher the predictability of a time series is the more recurrent are its temporal dynamics, i.e. the easiest will be to predict its future evolution.

δ-MAPS (Fountalis et al., 2018) is used to identify climate modes of variability. It is an unsupervised network analysis method that identifies spatially contiguous and possibly overlapping regions referred to as domains, and the lagged

functional relationships between them. This dimensionality reduction method is simpler and easier to interpret than empirical orthogonal functions (EOFs) which suffer from orthogonality constraints (Dommenget and Latif, 2002). Its benefits include interpretability and overfitting prevention relative to conventional EOF-based approaches when extracting climate patterns from high-dimensional datasets (Falasca et al., 2019). In short, δ-MAPS domains are spatially contiguous regions that share a highly correlated temporal activity between grid cells. In this work we apply it to the sea surface temperature (SST)

anomaly field (see Data) to identify major modes of climate variability in the north Pacific in a reanalysis and in the ESMs. Given any spatio-temporal field, its local homogeneity is hypothesized to be highest at "epicenters" or "cores". For each grid point, a local homogeneity is defined as the average pairwise cross-correlation between that grid cell and a set of *K* nearest

neighbors (see Fountalis et al., 2018 for details). Cores are then determined as neighbors of points where the local homogeneity is a local maximum and above a threshold δ. Each core is iteratively expanded and merged using a greedy algorithm to iteratively find domains as large as possible that are (i) spatially contiguous, (ii) include at least a core and (iii) have homogeneity higher than δ. δ is computed using a significance test for the unlagged cross-correlations. Given any random pair of grid points, the significance of the Pearson's correlation of their timeseries is assessed through the Bartlett's formula (Box et al., 2011) with the null hypothesis of no coupling. The significance of each correlation is tested for a user-specified significance level α, and δ is computed as the average of significant correlations. Here, we applied δ-MAPS with $K$ = 8 and α = 0.01.

Lastly, we adopt the approach introduced by Diffenbaugh and Giorgi (2012) (which builds on Williams et al., 2007, Diffenbaugh et al., 2008 and references therein) to identify hotspots of change. The SED is a non-parametric method, meaning it does not assume a specific probability distribution for the data. This flexibility makes it applicable to a wide range of datasets, regardless of their underlying distribution. The reader is referred to Turco et al. (2015) for its application to global atmospheric data. Here we apply to O2 and IPV* the procedure described in Turco et al. (2015), as follows. Hotspots are quantified through a Standard Euclidean Distance index (SED) that aggregates the changes in means, variability and extremes of the given spatio-temporal field according to:

$$SED = \sqrt{\sum_{i=1}^{N\Delta} \sum_{j=1}^{4} \left( \frac{\Delta_{ij}}{p95(|\Delta_{ij}|)} \right)^2} \qquad (3)$$

We compute in each grid point two SED indices, separately for $O_2$ and IPV*. $N\Delta$ is the total number of indicators per each variable and $i$ the index identifying each indicator, $j$ spans the seasons, so that $\Delta_{ij}$ is the $i^{th}$ indicator in the $j^{th}$ season, and p95 is the 95[th] percentile. The indicators and SEDs are computed point by point, i.e. each grid point has one value, therefore the percentile is computed spatially over all the grid points. Here we consider December-January-February as (boreal) winter, March-April-May as spring, and so on. We consider three indicators for each variable, evaluating changes in means, variability and extremes between two periods of equal length. *Period 1* covers 1950-1981 (1960-1986 for reanalysis and E3SM-2G ocean hindcast), and *Period 2* 1983-2014 (1988-2014 for reanalysis and hindcast) over the historical time, and 2036-2067 and 2069-2100 for the projected future. In equation (3) indicators of both periods are normalized on the 95[th] percentile calculated over *Period 1*, to fairly compare changes of hotspots intensity over time. We chose not to compare 2069-2100 with 1950-1981, but with 2036-2067 instead, because we want to track changes in each period compared to the preceding timeslot to quantify how rapidly they occur in the future projections compared to the historical time. For each variable, we compute three indicators at each grid point and for each season using the Climate Data Operators (Schulzweida, 2022) as follows:

- Changes in means are estimated in each season separately by $\Delta_{means} = yseasm_2 - yseasm_1$, where $yseasm_1$ and $yseasm_2$ are multi-year seasonal means in *Period 1* and *2*, respectively. Therefore, taking for example the $O_2$ historical simulations over 1950-2014 (but similar expressions hold for IPV* and the other periods), $\Delta_{means}\text{DJF} = <O_{2DJF}>_{1983-2014} - <O_{2DJF}>_{1950-1981}$, $\Delta_{means}\text{MAM} = <O_{2MAM}>_{1983-2014} - <O_{2MAM}>_{1950-1981}$, $\Delta_{means}\text{JJA} = <O_{2JJA}>_{1983-2014} - <O_{2JJA}>_{1950-1981}$ and $\Delta_{means}\text{SON} = <O_{2SON}>_{1983-2014} - <O_{2SON}>_{1950-1981}$, where $< ... >$ is a time mean (seasonal climatology).

- Changes in multi-year seasonal variability $\Delta_{variability}$ are evaluated by (i) detrending each variable point by point in the two periods separately, (ii) computing the multi-year seasonal standard deviation of these detrended fields, $yseas\sigma$, for each period for each season, (iii) computing as the percentual changes such that $\Delta_{variability} = 100 \cdot \left( \frac{yseas\sigma_2 - yseas\sigma_1}{yseas\sigma_1} \right)$. Therefore, with the example of $O_2$ historical simulations over 1950-2014, $\Delta_{variability}\text{DJF} = 100 \ (std(O_{2DJF})_{1983-2014} - std(O_{2DJF})_{1950-1981})/std(O_{2DJF})_{1950-1981}$, where $std(...)$ is the multi-year seasonal (winter) standard deviation over the specified periods (and equivalent formulations hold for the other seasons).

- Finally, changes in extremes (in our case overshoots of IPV* and undershoots of $O_2$) are computed through the following steps: (i) for each season, we compute at each grid point the multi-year $O_2$ minimum or IPV* maximum over *Period* 1 using monthly data (for example the $O_2$ minimum given all December, January and February values for boreal winter), and we build a threshold map for each season; (ii) we count how many times $O_2 < threshold_{O2}$ ( or IPV* > $threshold_{IPV*}$) is verified in each corresponding season of *Period 2* again using monthly data; (iii) the percentage of occurrences computed at point (ii) is finally taken as indicator of percentual changes in extremes and estimated by $\Delta_{extremes} = 100 \cdot \left( \frac{N_{occ}}{N_T} \right)$, where $N_{occ}$ is the number of extremes occurrences (in each season) and $N_T$ is the total number of months in the corresponding seasons (96 for the models and 81 for reanalysis and hindcast).

Building on previous works (Falasca et al. 2019; Falasca et al. 2022), we expect the statistics of a given model to remain relatively stable across ensemble members, i.e. we do not expect the member choice to significantly influence the calculation of extremes and hotspots and especially their relationships. We verified that this is indeed the case in one of the models by testing four additional randomly chosen ensemble members of CanESM5 (see Data). A major advantage of this hotspot definition is that it accounts for changes in mean, variability and extremes at the same time. In other words, it accounts for the intrinsic characteristics of the simulated climate fields, which can be characterized by considering all the three aspects together. In particular, the definition of extremes aims at including information on months exceeding corresponding seasonal baseline extremes, without choosing a priori a threshold on the current distribution, which is especially relevant for comparing changes with respect to a reference baseline. The three indicators, grouped into four seasons for each variable, are then used to compute the SED indices.

Finally, the IPV* ($m^{-1}$ $s^{-1}$) is used as a proxy of stratification and is defined as the isopycnic potential vorticity (Talley et al., 2011) with the absolute value of the Coriolis parameter in its formula:

$$IPV* = \frac{|f|}{g} N^2 \qquad (4).$$

Here $N^2$ is the Brunt–Väisälä frequency ($-\frac{g}{\rho}\frac{\partial\rho}{\partial z}$), which measures the fluid stability to vertical displacements, g is the gravitational acceleration, f is the Coriolis parameter and ρ is density, calculated in this work using salinity and temperature fields and the TEOS-10 equation of state for seawater (http://www.teos-10.org/). IPV is a conservative tracer in frictionless and adiabatic circulation. IPV* is calculated over the three-dimensional ocean volume using Eq. 4 considering the 0-200 m vertical weighted average. This procedure allows us to compare datasets with different vertical discretization.

## 3 Data

We consider four ESMs from the CMIP6 historical catalogue (with prescribed $CO_2$ concentrations), a hindcast and reanalysis data as summarized in Table 1. Whenever multiple ensemble members were available, we selected the first (r1i1p1f1). We randomly selected four additional ensemble members for CanESM5 (r5i1p1f1, r10i1p1f1, r15i1p1f1 and r20i1p1f1) to

220 further verify the robustness of the hotspots calculation to the member choice. All ESMs are forced with prescribed $CO_2$ concentrations from 1850 to 2014 and we analyse the monthly outputs from 1950 to 2014. We further discuss future ssp585 scenarios and focus on the 2036-2100 period, indicated as *future*.

The hindcast is a new ocean-ice biogeochemistry simulation (referred to as the G-Case), E3SMv2.0-BGC (hereafter, E3SM-2G, Takano et al., 2023), based on the Model for Prediction Across Scales-Ocean (MPAS-O), an ocean component of the

225 Energy Exascale Earth System Model (E3SM) version 2 (Golaz et al., 2022). Details on ocean physics updates can be found in Golaz et al. (2022). One of the major updates is the introduction of Redi isopycnal mixing (Redi, 1982). Along with ocean physics updates, this version also incorporated a uniform background vertical diffusion specifically developed for simulations of the ocean biogeochemistry to enhance ocean carbon uptake and thermocline ventilation of dissolved inorganic carbon (DIC). Incorporating this mixing parameterization results in an improved representation of climatological $O_2$

distributions in the v2.0 version compared to its predecessor (Burrows et al., 2020). The Marine Biogeochemistry Library (MARBL, Long et al, 2021) is used to simulate the ecosystem dynamics and cycling of biogeochemical elements. After the spin-up period, the model is forced by a meteorological reanalysis dataset, JRA-55do version 1.4 (Tsujino et al., 2020) from 1958 onward. As ocean reanalysis, we use the ORAS4 product (Balmaseda et al., 2012; Mogensen et al., 2012) available from 1959 onward, which includes a direct surface fluxes implementation from ERA40 and ERA-Interim and multi-scales

bias correction. When analyzing the E3SM-2G hindcast and the ORAS4 reanalysis, we focus on the 1960-2014 interval to avoid the spurious presence of an anticyclonic tropical cyclone in the NE Pacific in 1959 in JRA-55do v1.4.

All the data, models and reanalysis alike, are remapped at 1°x1° horizontal resolution and to a common vertical grid with a linear interpolator.

 **Table 1.** CMIP6 Earth System Models, global ocean hindcast and reanalysis used in this work. Under each model name a short form used in the following figures is indicated in parenthesis.

| Modeling Group/Center | Model Name | Atmospheric Component/Resolution | Oceanic Component/ Resolution | Reference |
|---|---|---|---|---|
| **National Oceanic and Atmospheric Administration, Geophysical Fluid Dynamics Laboratory** | GFDL-ESM4 (GFDL) | AM4.0, ~1°, 49 levels | OM4 MOM6, 0.5°x0.5°, 75 vertical levels (hybrid pressure/isopycnal)<br><br>Biogeochemical component:<br><br>COBALTv2 | (Dunne et al., 2020; Stock et al., 2020) |
| **Canadian Earth System Model version 5** | CanESM5 (CanESM) | CanAM5, T63 (~2.8°), 49 levels | CanNEMO, 45 vertical levels, NEMO3.4.1, ORCA1 tripolar grid, 1° with refinement to 1/3° within 20° of the Equator<br><br>Biogeochemical component:<br><br>CMOC | (Swart et al., 2019; Christian et al., 2022) |
| **NorESM Climate modeling Consortium** | NorESM2-LM (NorESM) | CAM-OSLO, 2° resolution; 32 levels. | MICOM, 1°, 70 vertical levels<br><br>Biogeochemical component:<br><br>iHAMOCC | (Seland et al.,2020; Tjiputra et al., 2020) |
| **Institut Pierre-Simon Laplace** | IPSL-CM6A-LR (IPSL) | LMDZ, NPv6, N96; 1.25°Lat x 2.5° Lon, 79 levels | NEMO-OPA (eORCA1.3, tripolar primarily 1°, 75 vertical levels. | (Boucher et al., 2020) |

| | | | Biogeochemical component: PISCESv2 | |
|---|---|---|---|---|
| **Department of Energy, Energy Exascale Earth System Model** | E3SMv2.0-BGC (E3SM) | JRA55do reanalysis (55km, 3hr resolution) | MPAS-O (30 to 60km resolution)<br><br>Biogeochemical component:<br><br>E3SMv2.0-BGC, MARBL | (Golaz et al, 2022; Takano et al., 2023; Long et al., 2021) |
| **ECMWF Ocean reanalysis System** | ORAS4 | _ | Global, 1°, 42 Levels | (Balmaseda et al.,2012; Mogensen et al.,2012) |

We begin our analysis with a brief evaluation of the ESM biases in the two main fields of interest, IPV* and $O_2$. For the IPV*, the ocean reanalysis dataset is used for validating the model outputs for the maximum possible time overlap in the historical configuration (1959-2014). For $O_2$, we can only contrast the annual mean $O_2$ climatology between the World
Ocean Atlas (Garcia et al., 2019) and the ESMs (Fig. 1). We additionally compared the ORAS4 IPV* climatology over 1988-2014 (i.e. "period 2" for this reanalysis) with the corresponding climatology computed using SODA3.4.2, which uses a different ocean component than ORAS4. The differences across reanalysis that use different models but assimilate the same observations are much smaller (about one order of magnitude) than the signal (Suppl. Fig. S1), and smaller than any model bias.

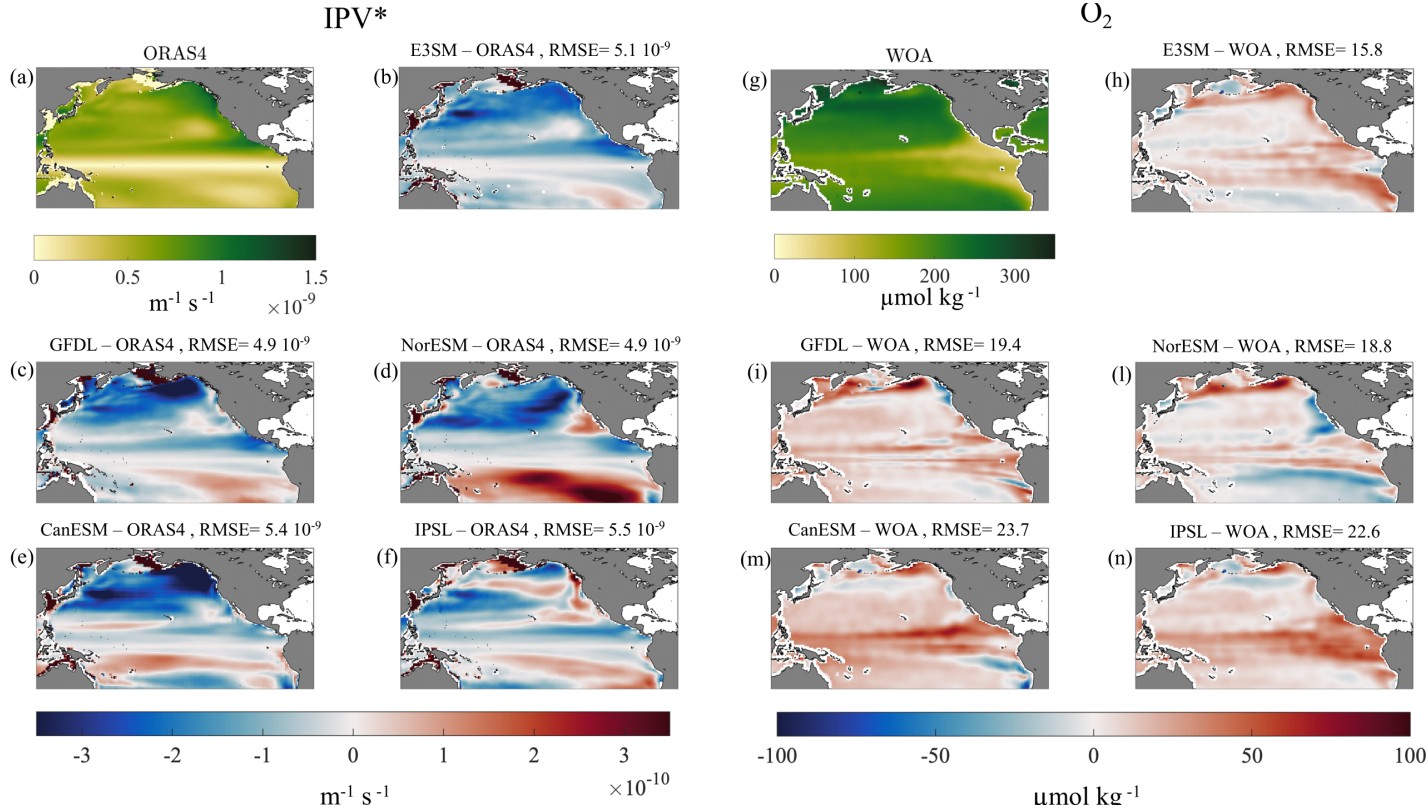

**Figure 1**. (Left) IPV* annual mean climatology (1959-2014) weighted averaged over 0-200 m depth in the North Pacific basin. (a) ORAS4. (b-f) Model biases (model – ORAS4) difference. (Right) $O_2$ annual mean climatology (1950-2014) weighted averaged over 0-200 m depth. (g) World Ocean Atlas climatology. (h-n) Model biases (model – WOA) difference. The Root Mean Squared Error (RMSE) of the modelled IPV* ($m^{-1} s^{-1}$) and $O_2$ (micro mol $kg^{-1}$) is shown for each panel.

The E3SM-2G hindcast is forced by observed atmospheric fields and displays the smallest bias and, for O2, also the smallest root mean square error (RMSE), which is shown atop of the panels in Figure 1. Overall, the IPV* and $O_2$ biases have broadly anticorrelated patterns, with the models being generally less stratified and more oxygen rich than observed in the extra-tropical North Pacific, and often too stratified and with a larger $O_2$ deficit than observed south of the Equator. However, maximum and minimum biases in the two fields only seldom coincide. Regionally, E3SM-2G is generally less stratified than observed with a relatively low $O_2$ bias and an overestimation of approximately 10 micro mol $kg^{-1}$ in the subtropical thermocline of the North Pacific basin. The hindcast performs especially well in the tropical thermocline. Among the CMIP6 models, CanESM5 shows a slightly higher IPV* underestimation in the subpolar gyre and a $O_2$ overestimation in the subtropics compared to the other ESMs, while NorESM2-LM emerges as the most stratified south of the Equator. For O2

larger biases (positive or negative) are found generally in the tropical thermocline and the tropical/subtropical boundaries. The sign and magnitude of the biases are model dependent. Interestingly, models generally overestimate $O_2$ at subpolar latitudes.

## 4 Results

### 4.1 Predictability potential (HYP 1)

We begin our analysis considering the predictability potential of IPV*, quantified through the information entropy (IE, see Methods). The goal is to verify if and where IPV* has an elevated predictability, owing to the presence of quasi-recurrent behaviors in its time-series. We also aim to examine whether $O_2$ is correlated with IPV* in regions where the latter has a high predictability potential. As a reminder, IE values close to 1 indicate high complexity and unpredictability, and close to 0 perfect predictability (the signal is recurrent, for example constant or periodic). We preliminary tested the sensitivity of the entropy field to the microstate dimension, within a meaningful range according to previous literature (Ikuyajolu et al., 2021), using microstates of dimension 2, 3, 4 and 5 for GFDL over 1950-2014 (Suppl Fig. S2). We found that the IE pattern, i.e. areas more (less) predictable relative to the surroundings are substantially unchanged, and the geographical patterns are robust for changes in microstate dimension, in agreement with Ikuyajolu et al. (2021). Both microstate dimensions 4 and 5 show reasonable entropy values and we chose to use a microstate dimension of 4 to conduct our analysis because it spans the widest range of possible values.

### 4.1.1 $O_2$ – IPV* relationship across ESMs and its future evolution

IE maps for IPV* are shown in Fig. 2 for both historical and future times, with superposed the contours of the areas where the (lagged) anticorrelation between IPV* and $O_2$ is at least -0.5 (see Suppl. Fig. S3-S4 for the anticorrelation and lag maps). Higher predictability in the historical period is found in the tropical Pacific areas close to the geographical location of ENSO (i.e. the area most impacted by the domain identified as ENSO-related by δ-MAPS, which well maps the region identified by an EOF analysis over the SST field for having the greatest variance explained by the first principal component, *pc1*). The predictability potential is generally highest along two stripes enclosing the ENSO pattern north and south of the Equator and excluding the upwelling cold tongue. The distribution of IE follows broadly that found in a much longer simulation of the IPSL model covering the past 6,000 years and analyzed by Falasca et al. (2022) and appears to be robust across models. The western boundary current region and the Kuroshio-Oyashio extensions have low predictability across all datasets considered. In NorESM2-LM and CanESM5, and to a lesser degree in ORAS4 and IPSL-CM6A-LR, the higher predictability of the ENSO area extends to the north-eastern portion of the basin. In general, in both the hindcast and the models, strong anticorrelations between IPV* and $O_2$ (c.c $\leq$ -0.5) coincide with low IE regions and are linked to ENSO affecting concurrently stratification and $O_2$ in the tropics and south of the upwelling area. Very limited IPV* predictability is found in

the central and western North Pacific, where the variability is dominated by the PDO signal. The PDO does not emerge as easily predictable in the interval considered, in agreement with e.g. Gordon et al. (2021) and Falasca et al. (2022) who analyzed predictability of sea surface temperatures in the IPSL model across the whole second half of the Holocene. In those areas, anticorrelations between $O_2$ and IPV* are relatively weak (generally > -0.4 but for NorESM). The entropy and the

regions where the evolution of IPV* and $O_2$ are strongly anticorrelated do not change significantly in the future projections in the four models. We further explored whether oxygen solubility, ($O_{2sol}$), which is modulated by ocean warming/cooling, and the apparent oxygen utilization AOU, which is controlled mostly by the biogeochemical processes affecting oxygen demand, may be independently linked to IPV* predictability. The areas where IPV* and AOU time series are positively correlated with correlation coefficients $\geq 0.5$ are very similar to the ones obtained by analyzing the $O_2$-IPV* relationship. For

$O_{2sol}$, which well approximates preformed $O_2$ at the depths considered, the anticorrelations areas (i.e. where c.c. $\leq$ -0.5) are quite extensive, especially in the hindcast, but mostly superimposed to high-entropy/low predictability IPV* areas (Suppl. Fig. S5).

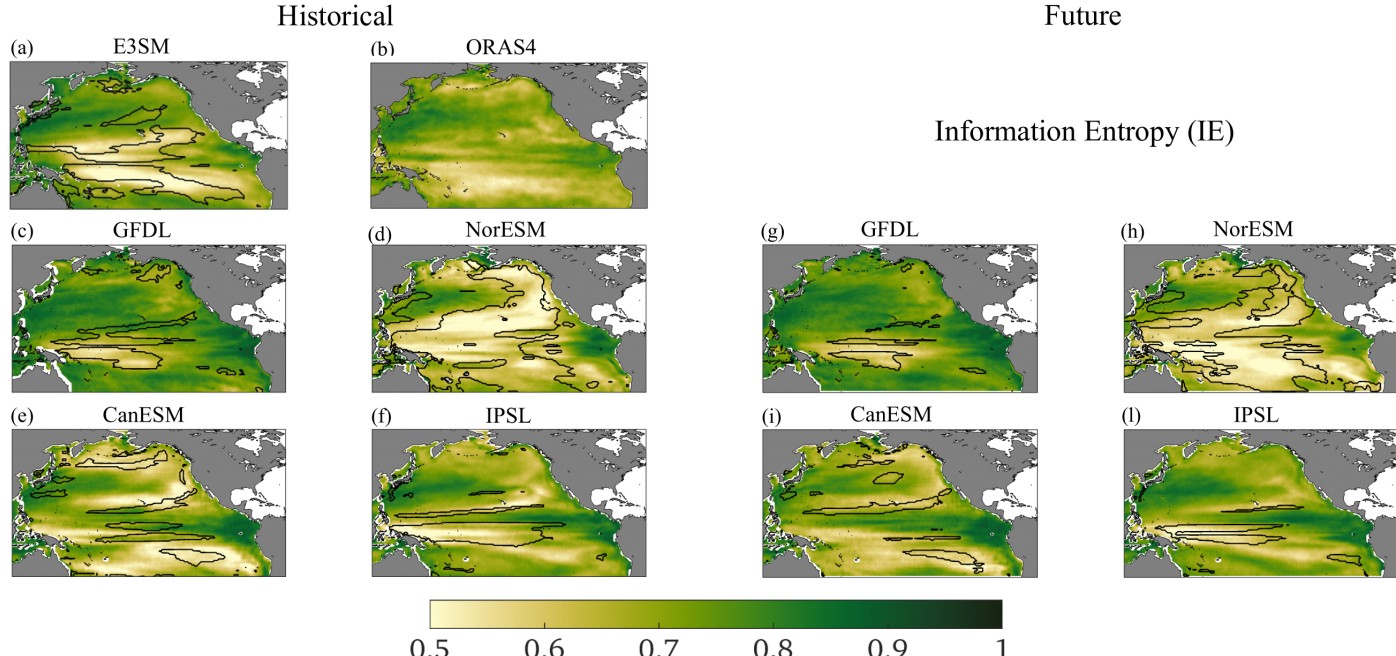

**Figure 2**: IPV* information entropy in the historical interval (left) and in the future (right) for the ESMs, and in the historical 1960-2014

period for the hindcast and ORAS4, with superposed the contours of the areas where IPV* and $O_2$ time series are anticorrelated with correlation coefficients $\leq$ -0.5.

In Suppl. Fig. S6 we show the IE maps for $O_2$. Predictability is generally higher outside the equatorial upwelling region and the Kuroshio extension. In E3SM the areas where the predictability potential of IPV* is high and the correlation with $O_2$ is $\leq$

0.5 coincide and have also low IE in $O_2$. The same is somewhat verified in IPSL and NorESM but not in the other two models.

In the next section we isolate the PDO signal to explore whether the low predictability in the northern Pacific Ocean (north of ENSO region) is related to the superposition of different time scales, i.e. if there is a low frequency PDO modulation with a high frequency "noise" due to both atmospheric and oceanic variability. The PDO is indeed a lower-frequency mode compared to ENSO and has most loading at extra-tropical latitudes where the atmospheric "noise" is greater.

## 4.2 Trends and PDO impact on $O_2$ and IPV* (HYP 2)

The limited predictability found in the North Pacific does not exclude the possibility of the PDO modulating both IPV* and $O_2$ inventories simultaneously. As explored in previous works (for example, Ito et al., 2019), the dominant mode of observed $O_2$ variability in the northern Pacific Ocean is indeed correlated with the PDO index which explains about 25% of the variance. Observations, however, offer only sparse coverage, in both time and space. To further verify the PDO modulation, we computed the first EOF for the E3SM-2G hindcast 0-200 m $O_2$ and IPV* anomalies over 1960-2014 over the northern Pacific (20.5°N-69.5°N;115.5°E-60.5°W) and the corresponding time series for *pc1*. The first EOF explains 25% of the oxygen variance and about 12% of the IPV* variance. The computed *pc1* shows a significant and strong correlation (Pearson's R coefficient) with the PDO timeseries computed using SST anomalies with $|R| = 0.83$ ($p < 0.01$) for $O_2$, and $|R| = 0.44$ ($p < 0.01$) for IPV*, after applying 5-year moving means. The correlation with the PDO is higher than with ENSO, which is at most $|R| = 0.34$ ($p < 0.01$) for $O_2$, after applying a 3-month moving mean, consistent with analysis by Ito et al. (2019). We hereby quantify the (linear) impact of the PDO on the two fields of interest, and then evaluate their residuals. If the PDO is the main predictor of IPV* and $O_2$ distributions, its impact on the two fields should be strongly anticorrelated and larger than the residual. As mentioned, the objective is to verify if IPV* could be used to extrapolate information about $O_2$ and its evolution in time, bypassing – at least to some degree – the need to run full biogeochemical models or measure $O_2$ directly.

### 4.2.1 Estimation of climate indices and their relationship with IPV* and O2

We use δ-MAPS (see Material and Methods) applied to the SST field to evaluate the main modes of Pacific climate variability, ENSO and PDO, and their time evolution in the models, the ocean hindcast and the reanalysis. While the evolution of ENSO using δ-MAPS is straightforwardly described by the timeseries of the cumulative anomalies in the ENSO-related domain (e.g. Falasca et al., 2019), for the PDO we must consider the difference between the SST cumulative anomalies of two domains. The domains are identified by the complex network algorithm, and we applied a 5-yr running mean to produce the PDO timeseries shown in Fig. 3. The domains shape and size are indicated in Fig. 4. For ORAS4 and E3SM-2G over 1960-2014, we computed the 0-lag Pearson's correlation coefficients between these timeseries and the

commonly defined indices of PDO (following Mantua et al. 1997) and Nino34 (average SST anomalies over the box 5°N-5°S, 170°W-120°W) retrieved from NOAA (https://psl.noaa.gov/data/climateindices/list/). After applying a 3-month moving average to the ENSO timeseries (signals and indices) and a 5-year moving average to the PDO timeseries (signals and indices), the correlation coefficients are 0.88 for PDO and 0.93 for ENSO in ORAS4, and are 0.89 for PDO and 0.91 for ENSO in E3SM-2G.

Among the models (Fig. 3), in the historical period GFDL slightly underestimates the PDO strength, while the opposite is verified in CanESM and NorESM. In the latter, the frequency of the signal is also higher than observed. By the end of the 21st century, the strength of the PDO remains unaltered in GFDL and IPSL, while decreases in NorESM2 and especially in CanESM, following a decrease in size of the eastern domain. A decrease in amplitude and increase in frequency of the PDO was found also in several models in the CMIP5 ensemble (Li et al., 2020).

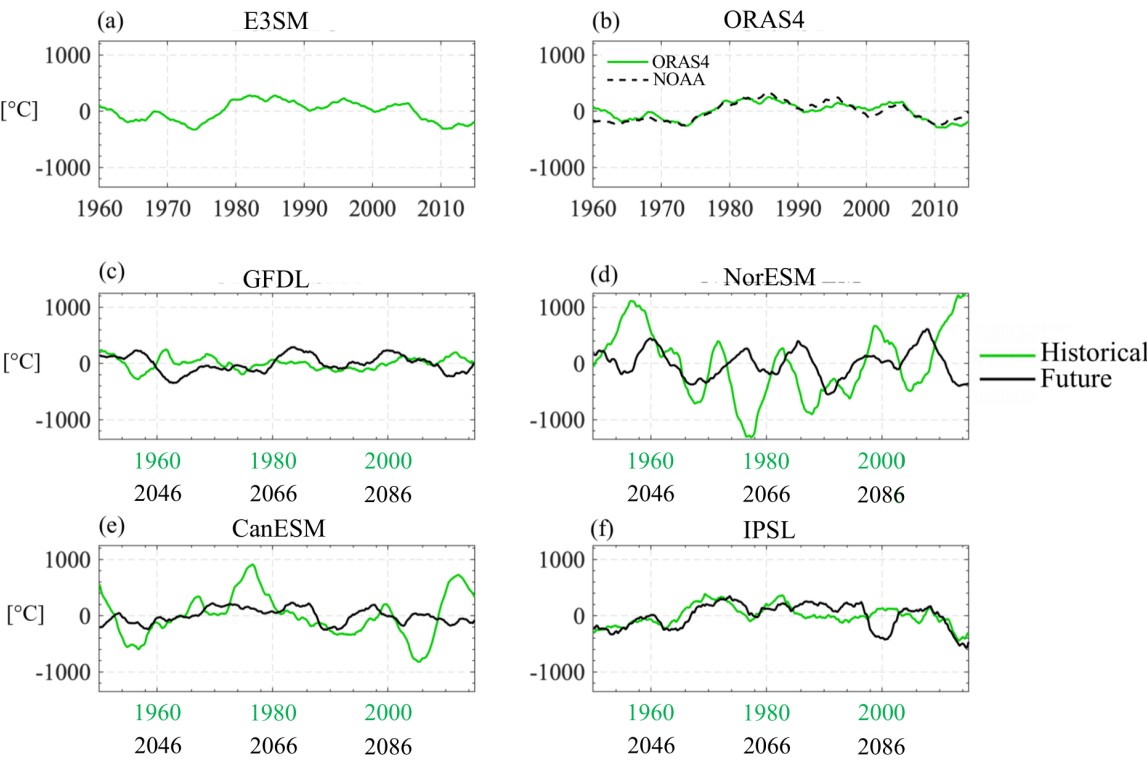

Figure 3: PDO indices (SST cumulative anomalies) calculated using δ-MAPS (see text) in the historical and future time periods. The dashed line in panel (b) is the PDO index timeseries from NOAA (available at https://psl.noaa.gov/data/climateindices/list/) multiplied by the standard deviation of the ORAS4 timeseries in panel (b) after applying a 5-year moving mean.

Given the PDO(t) indices, the residual component of the fields of interest that is not linearly forced by the PDO can be separated as a function of time (see e.g. Kucharski et al., 2008) so that for $O_2$ (but the same procedure was applied to IPV*):

$O_{2res}(x,y,t)=O_2(x,y,t)-O_{2PDO}(x,y,t)$ (5),

where

$O_{2PDO}(x,y,t) = b_{O2}(x,y)*PDO(t)$ . (6)

$b_{O2}(x,y)$ is constant in time and determined by least-square fitting through a linear regression for each dataset separately. Figure 4 shows $b_{IPV*}$ and $b_{O2}$ for all datasets with superposed the boundaries of the domains corresponding to the ENSO

mode and those contributing to the PDO in the historical period. In most cases there is an overall anticorrelation between the maps of the two fields, but also several important differences. First, the regions where $b_{O2}$ is strongest (both positive and negative values) do not correspond to minima and maxima in $b_{IPV*}$. Second, the equatorial upwelling tends to have a strong positive signal in $b_{O2}$ and only a weak one, but of the same sign, in $b_{IPV*}$. Third, the PDO impacts on the fields vary substantially among models, as quantified by the correlation among the respective fields indicated atop the $b_{O2}$ plots, with

GFDL being the closest to the hindcast and, for the IPV* case, also to the reanalysis. In NorESM2 the anticorrelation between the regression fields is too strong and the PDO has both shape and loading different than observed in the Pacific interior. CanESM and IPSL display positive spatial correlations, with important biases with respect to the hindcast at the equator and along the eastern boundary.

We performed a comparable linear regression analysis using the ENSO index, instead of the PDO, and obtained similar

shapes of the b coefficients as expected, but much lower absolute values (Suppl. Fig. S8). This further confirms that in the North Pacific the PDO is the dominant mode of climate variability.

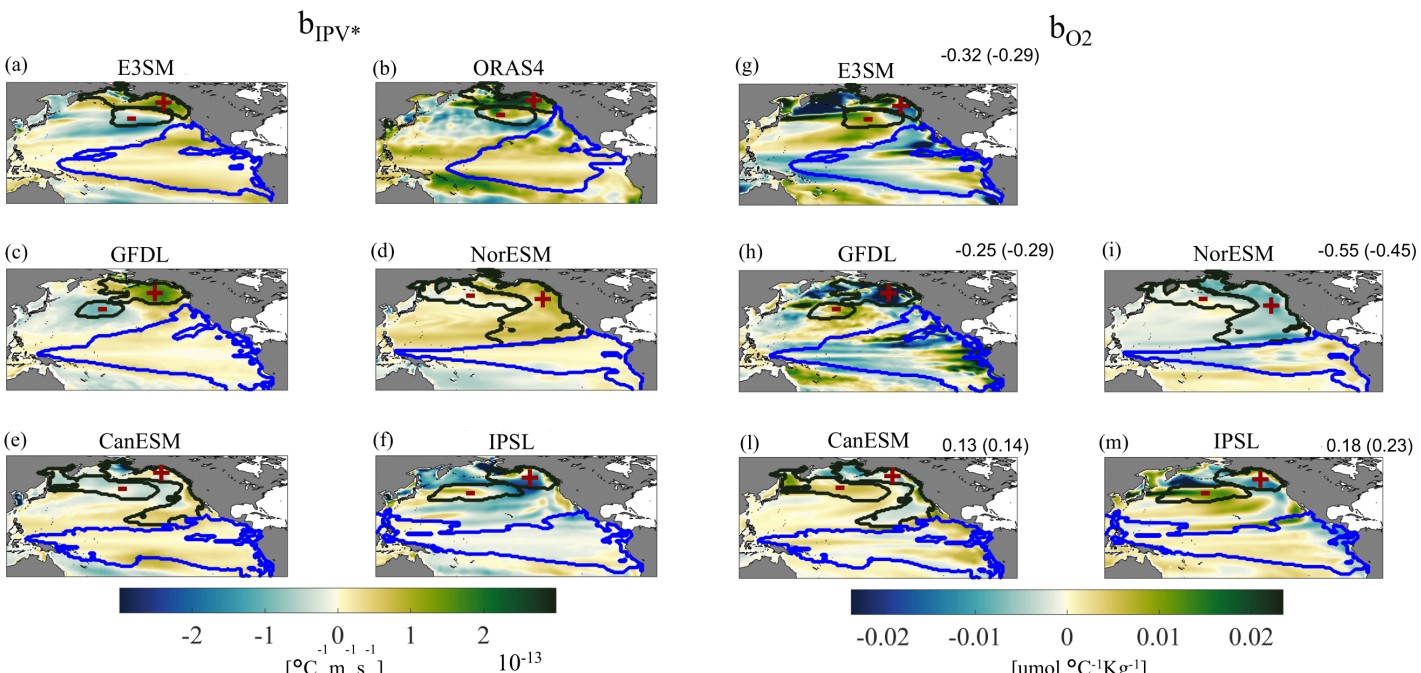

$b_{IPV*}$             $b_{O2}$

**Figure 4**: $b_{IPV*}$ (left) and $b_{O2}$ (right) regression coefficient maps with superposed contours of the ENSO (blue line) and of the PDO+ and PDO- domains. $b_{IPV*}$ represents the change in IPV* per unit change in SST, $b_{O2*}$ represents the change in $O_2$ per unit change in SST. The correlation coefficients (c.c.) among the corresponding maps for the same model or hindcast are also indicated. Color limits are fixed as +/- 3 standard deviations of the ensemble for each variable over the whole area (+/- 2.99 $10^{-13}$ for IPV* and +/- 0.023 for $O_2$). Values in parentheses are c.c. computed north of 20°N. All the c.c. values passed a shuffling significance test at 5% level (see Suppl. Mat.).


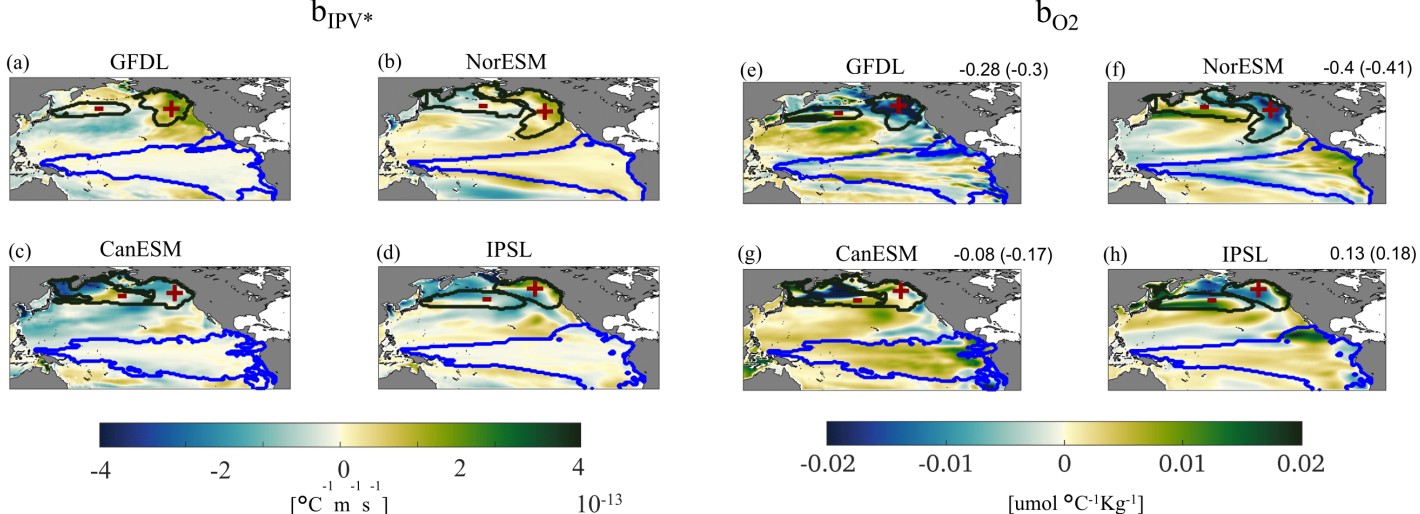

**Figure 5:** as in Figure 4 but for the future projections. Color limits are fixed as +/- 3 standard deviations of the ensemble for each variable over the whole area (+/- 4.1 $10^{-13}$ for IPV* and +/- 0.02 for $O_2$). Values in parentheses are correlation coefficients (c.c.) computed north of 20°N. All c.c. passed a significance test at 5% level (see Suppl. Mat.).

Moving to the projections, the maps of the regression coefficients do not change considerably in three of the models considered (Fig. 5). In CanESM, on the other hand, $b_{IPV*}$ changes sign over most of the domain. The residual trends, when compared to the regression coefficients, are stronger and dominate the evolution of both fields, especially in the subtropical and subpolar gyres of the North Pacific (Suppl. Fig. S9), superseding the PDO signal.

Overall, in the historical period the residuals have amplitude comparable to the PDO-forced signal (see Suppl. Fig. S7), and the linear trends have similar patterns to but less amplitude than the whole fields trends (see further discussion of trends shape when presenting $\Delta_{means}$). In the future, on the other hand, linear trends of residuals and whole fields have similar patterns and intensities.

## 4.3 Hotspots of change (HYP 3)

As a last step, we evaluate changes in means, variability and extremes for both variables (considering whole signals, i.e. not just the residuals) using the indicators introduced in the Methods. For the historical time, we divide the 1950-2014 interval in two periods of equal length covering 1950-1981 and 1983-2014 (1960-1986 and 1988-2014 for E3SM-2G and ORAS4). We evaluated the indicators in each season separately or averaged together, and found that differences across seasons were small,

as measured by the standard deviation of the indicators (Suppl. Fig. S10-S12). In the following we discuss only the indicators averaged across the four seasons without loss of information.

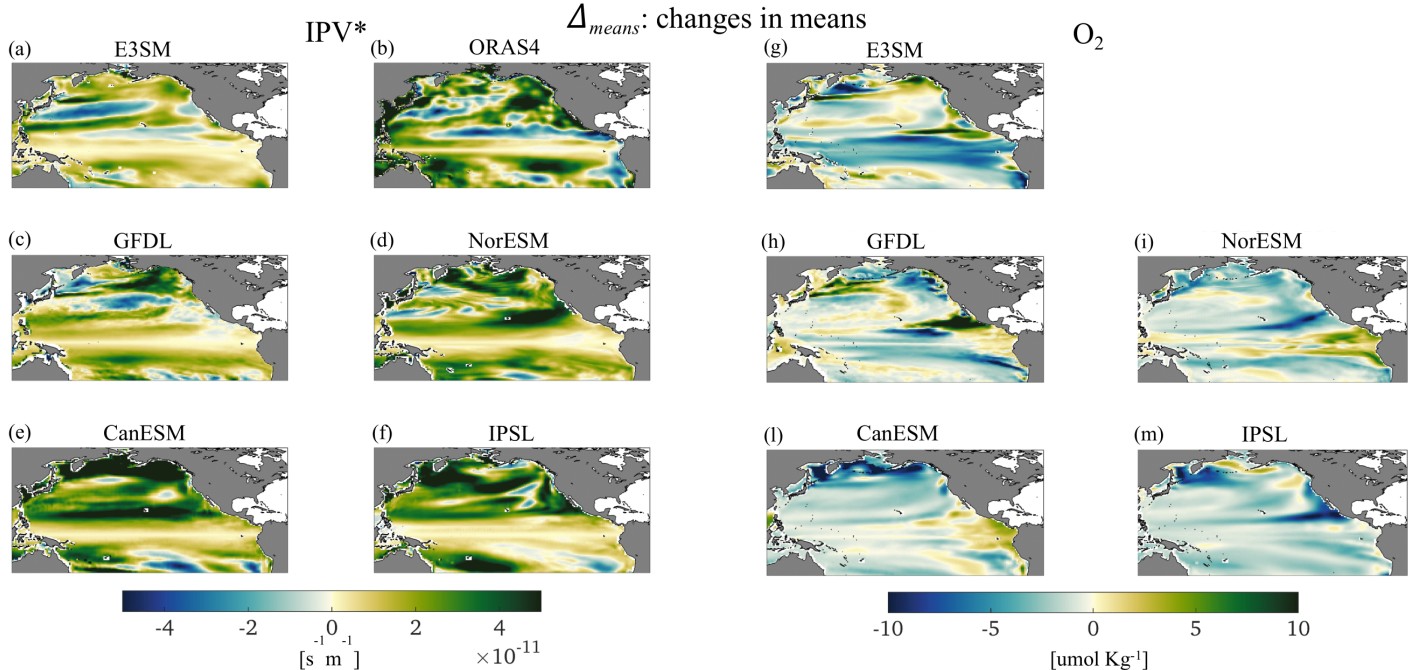

**Figure 6.** 1950-2014 $\Delta_{means}$ for IPV* (left) and $O_2$ (right). All indicator maps are obtained by averaging the respective seasonal maps.

$\Delta_{means}$ in Fig. 6 shows the changes in the mean fields, which have very similar patterns to the linear trend in both IPV* and $O_2$ (see Suppl. Fig. S7). By 2015 stratification has increased nearly everywhere in the ESMs, but for the equatorial upwelling region, and the Kuroshio-Oyashio extension. In ORAS4 there is also a prominent band where stratification decreases between 10° and 20°N from the coast of the American continent to 150°W in the second period and in the overall trend. $O_2$ decreases in most of the north Pacific, especially in the subpolar gyre around the Kamchatka peninsula, and increases in the

upwelling areas along the coast of Peru, Central America, and the California Current System. Regions of increasing $O_2$ are also found in correspondence of the North Equatorial Current in the E3SM-2G hindcast, and in the GFDL and CanESM models, in the Equatorial upwelling band in NorEMS2, and in portions of the subpolar gyre around Alaska in E3SM-2G and IPSL.

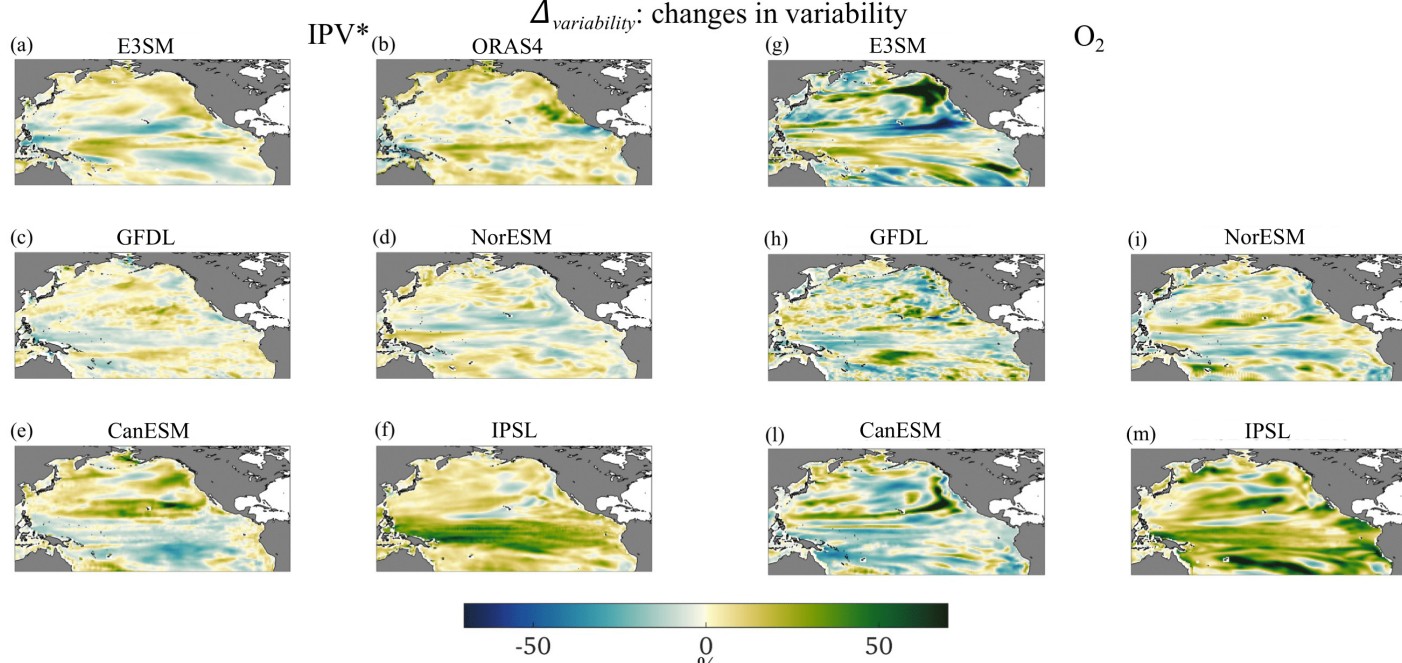

**Figure 7**. 1950-2014 $\Delta_{variability}$ for IPV* (left) and $O_2$ (right).

Indicators of change in (seasonal) variability ($\Delta_{variability}$, Fig. 7) show strong differences across models in patterns and, at least for $O_2$, intensity. Whenever corresponding maps of $O_2$ and stratification have the same sign and comparable amplitude at corresponding locations, they indicate that increments or decreases in IPV* variability at seasonal scales are associated with corresponding increments in 0-200 m $O_2$ variability. In the hindcast, changes are greater for residual $O_2$ than stratification. This is verified in three of the models in the north-eastern extratropics. Among the models, GFDL and NorESM2 show patchy changes, both positive and negative, across the domain, with the smallest amplitudes among the datasets considered. CanESM undergoes predominately positive changes north of the equator in IPV* and negative to the south of it, while the variability in the $O_2$ field decreases also in the central portion of the subtropical gyre. In IPSL the variability increases nearly everywhere in both fields, but especially at the equator and to the south of it in IPV* and more uniformly at all latitudes in $O_2$.

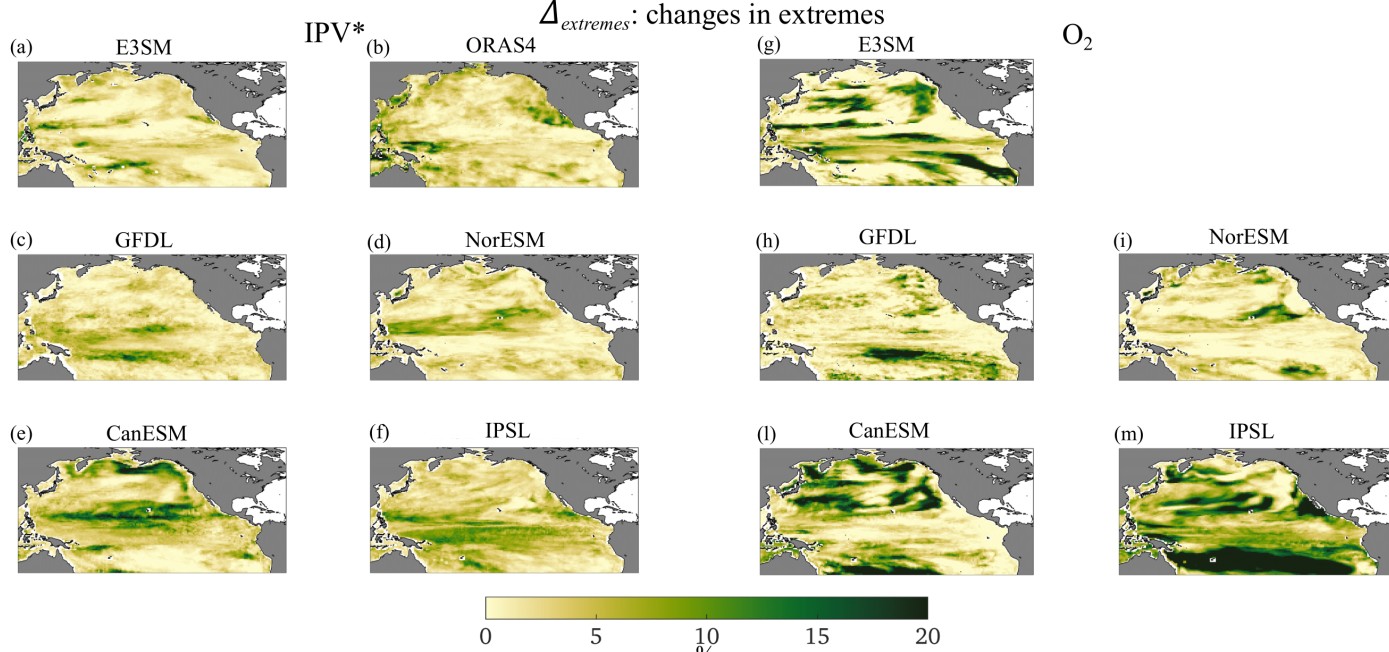

**Figure 8.** 1950-2014 $\Delta_{extremes}$ for IPV* (left) and $O_2$ (right).

Changes in extremes ($\Delta_{extremes}$) for the $O_2$ field are stronger than for stratification (Figure 8). Episodes of strong $O_2$ decrease and stratification increase are more frequent in *Period 2*. For $O_2$ the regions to the north and south of the equatorial upwelling band emerge as most impacted in the E3SM-2G hindcast and GFDL, while the subtropical gyre displays an increase in extreme events nearly everywhere in CanESM and IPSL, at its boundary in E3SM-2G, and in its eastern portion in GFDL and NorESM. The subpolar gyre is affected especially in CanESM and IPSL. Changes in IPV* extremes have less clear latitudinal differences and do not display a robust intensification at extratropical latitudes across the models. In ORAS4 maxima are found near the California Current System and in the Warm Pool area.

| | E3SM-2G | GFDL | NorESM | CanESM | IPSL |
|---|---|---|---|---|---|
| $\Delta_{means}$ (means) | -0.01 (-0.02) | -0.07 (-0.07**)** | **-0.23 (-0.16)** | **-0.28 (-0.16)** | **-0.12** (-0.07) |
| $\Delta_{variability}$ (variability) | **0.32 (0.24)** | **0.28 (0.23)** | 0.04 (**0.15**) | **0.33 (0.21)** | **0.24 (0.18)** |
| $\Delta_{extremes}$ (extremes) | 0.01 (-0.08) | **0.29** (0.05) | **0.35 (0.47)** | 0.09 (**0.34**) | -0.03 (**0.17**) |

**Table 2** 1950-2014 Correlation coefficients (c.c) between the corresponding indicator maps for IPV* and $O_2$. Bold underlined values indicate c.c. $\geq 0.1$ that passed the shuffling significance test at 5% level (see Suppl Mat.). Numbers in parentheses reflect c.c. computed north of 20°N.

Table 2 summarizes the correlation coefficients between the maps of the three indicators for the two fields considered. Coefficients are negative for all models but small for $\Delta_{means}$, slightly larger in amplitude and positive for the variability

indicator ($\Delta_{variability}$), and very small for $\Delta_{extremes}$ in the hindcast, CanEMS and IPSL, while larger in amplitude and positive for GFDL, with a strong contribution from the Equatorial region, especially in NorESM where positive values are also found north of 20°N.

The resulting hotspot indices (SED), computed separately for the IPV* and the $O_2$ indicators (see Methods) are reported in Fig. 9. Except for IPSL, the hotspots are found outside the equatorial band. Those for $O_2$ are generally stronger along the

eastern part of the subtropical gyres, in the eastern part of PDO region and along the California upwelling system, and the IPV* hotspots are more commonly found over the western parts of the basin and along the southern boundary of the subtropical gyre. This result suggests a longitudinal decoupling between hotspots in $O_2$ and stratification in at least three of the models and in the hindcast, with NorESM being the exception due to the simulated superposition of the changes in extremes in the two fields. We also computed the SED for the residual fields, obtaining similar results (Suppl Fig. S13).


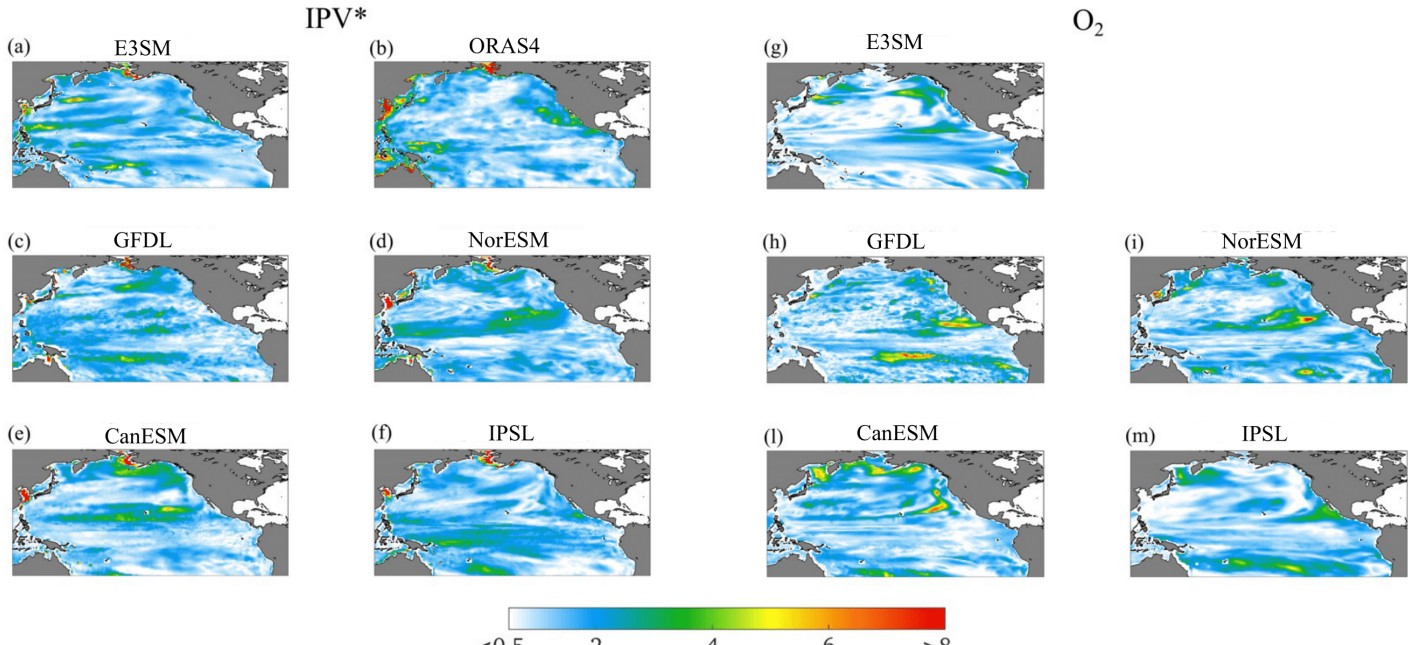

**Figure 9.** 1950-2014 (1960-2014 for E3SM-2G and ORAS4) SED *index* for IPV* (left) and $O_2$ (right). The colorscale is realized with rgbmap (Greene, 2023).

The maps of the indicators for the future projections follow in Fig. 10-12, again averaged over seasons, and the associated standard deviations are reported in Suppl. Fig. S14-S16. In the projections, the seasonal differences are slightly stronger compared to the historical period for $\Delta_{means}$ (IPV*) in the northern subpolar gyres especially for CanESM, NorESM and IPSL (Fig. 10), and along the subtropical and the northern subpolar gyres for $\Delta_{extremes}$ (IPV*) (Fig. 12). Standard deviations for $\Delta_{extremes}$ ($O_2$) are stronger along the extratropical gyres and weaker in the tropical upwelling region (Suppl. Fig. S16). Areas

of higher standard deviations in the projections are, however, associated with much stronger values of $\Delta_{means}$ and $\Delta_{extremes}$ compared to the historical period. In the projections, $\Delta_{means}$ strengthens significantly and is stronger than the actual trend shown in Supp. Fig. S9, indicating an acceleration of the changes in the last portion of the 21$^{st}$ century. This is especially relevant for IPV* north of the Equator. Stratification increases everywhere but for regions in the southern hemisphere with different extension in the four models and mostly located in the central and eastern portions of the basin. $O_2$ decreases

everywhere but for small areas around the equatorial upwelling band. The decrease is very strong along the northern boundary of the Pacific Ocean and, depending on the model, at the subtropical gyre boundary (NorESM and to a lesser degree CanESM) and south of the Equator along the coast of Central and South America (IPSL).

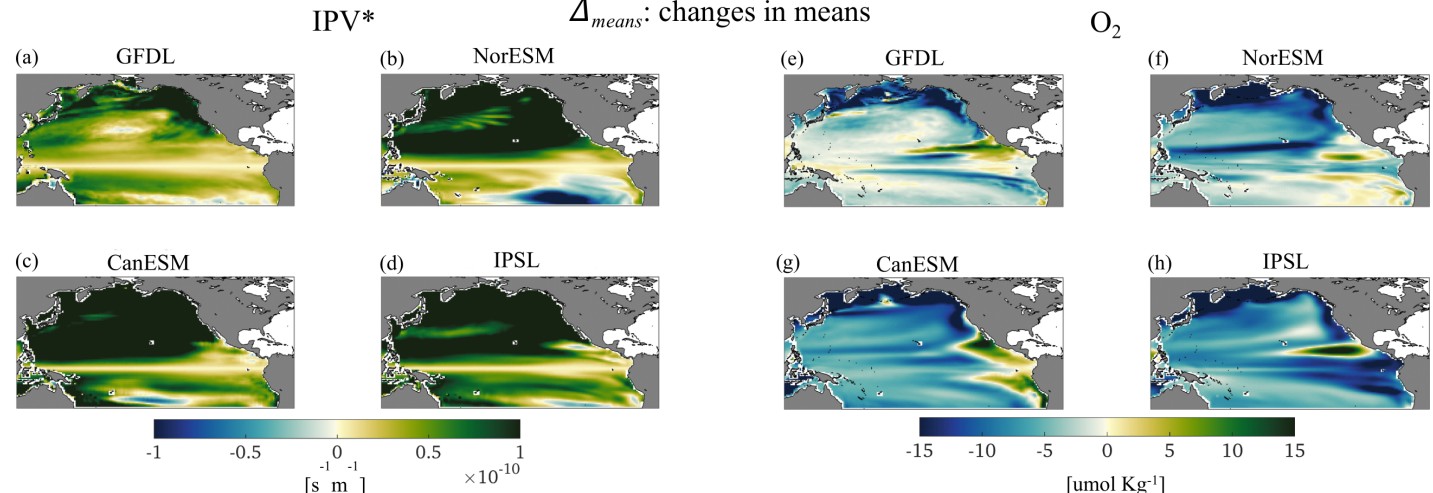

**Figure 10.** 2036-2100 $\Delta_{means}$ for IPV* (left) and O$_2$ (right).

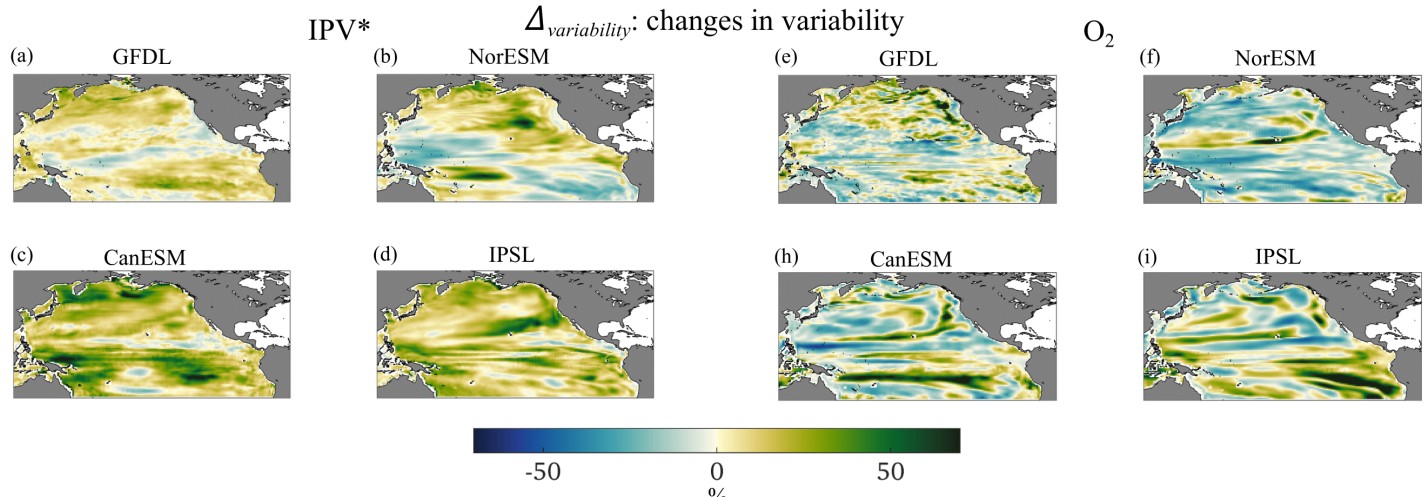

**Figure 11.** 2036-2100 $\Delta_{variability}$ for IPV* (left) and O$_2$ (right).

In terms of variability, when comparing the two variables very few areas in Fig. 11 have comparable sign and amplitude (which would indicate comparable increases or decreases). IPV* variability increases in the Warm Pool and to the south of the Equator in the eastern portion of the basin in all models but NorESM. O$_2$ variability increases in patchy areas mostly in the eastern half of the basin in GFDL, only along the southern boundary of the subtropical gyre in NorESM, roughly along the boundaries of the gyres in CanESM and along the northern gyre boundary and south of the Equator in IPSL. Lastly, the

extremes ($\Delta_{extremes}$, Fig. 12) increase in CanESM and IPSL nearly everywhere but for the equatorial upwelling area in both variables, in NorESM in the northern hemisphere for $O_2$ and in the ENSO region for IPV*, and in GFDL along the northern boundary of the basin for IPV* and in the northern and southern portion of the domain for $O_2$. Correlations among maps of the two variables are generally very small for all indicators in the projections (Table 3) with |c.c.| < 0.4, except for $\Delta_{means}$ in NorESM and CanESM. Finally, we verified the robustness of our results to the choice of the ensemble member, computing the extremes indicators of four randomly-chosen ensemble members of the CanESM model for the whole signals of IPV* and $O_2$ during the historical periods. We found no significant changes in extremes and SED (Suppl Fig S17-18).

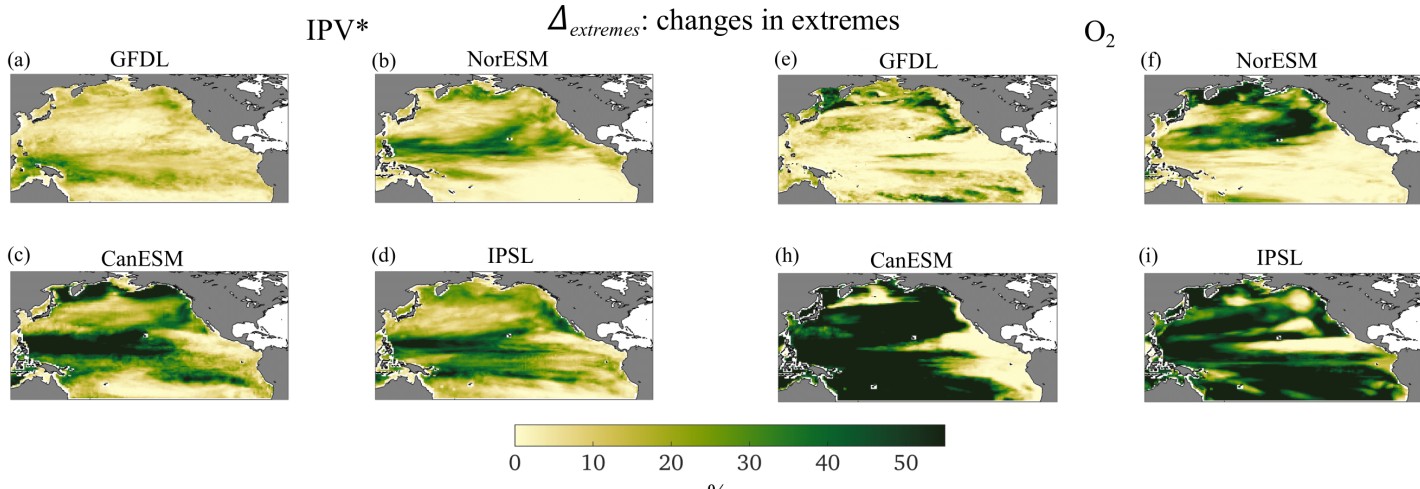

**Figure 12.** 2036-2100 $\Delta_{extremes}$ for IPV*$_{res}$ (left) and residual $O_{2res}$ (right). The percentage shown reaches 60% (three times more than during historical, Figure 8).

|  | GFDL-ESM4 | NorESM2-LM | CanESM5 | IPSL-CM6A-LR |
|---|---|---|---|---|
| $\Delta_{means}$ | **- 0.21 (-0.15)** | **-0.55 (-0.22)** | **-0.60 (-0.59)** | **-0.32 (-0.28)** |
| $\Delta_{variability}$ | **0.30 (0.30)** | **0.14 (0.26)** | **0.21 (0.19)** | 0.02 (0.00) |
| $\Delta_{extremes}$ | -0.01 **(0.56)** | **0.35 (0.40)** | **0.23** (0.01) | 0.08 (-0.02) |

**Table 3**. 2036-2100 Correlation coefficients (c.c) between the corresponding indicator maps for IPV* and $O_2$. Bold underlined values indicate c.c. $\geq 0.1$ that passed the shuffling significance test at 5% level (see Suppl Mat.). Numbers in parentheses reflects c.c. computed north of 20°N.

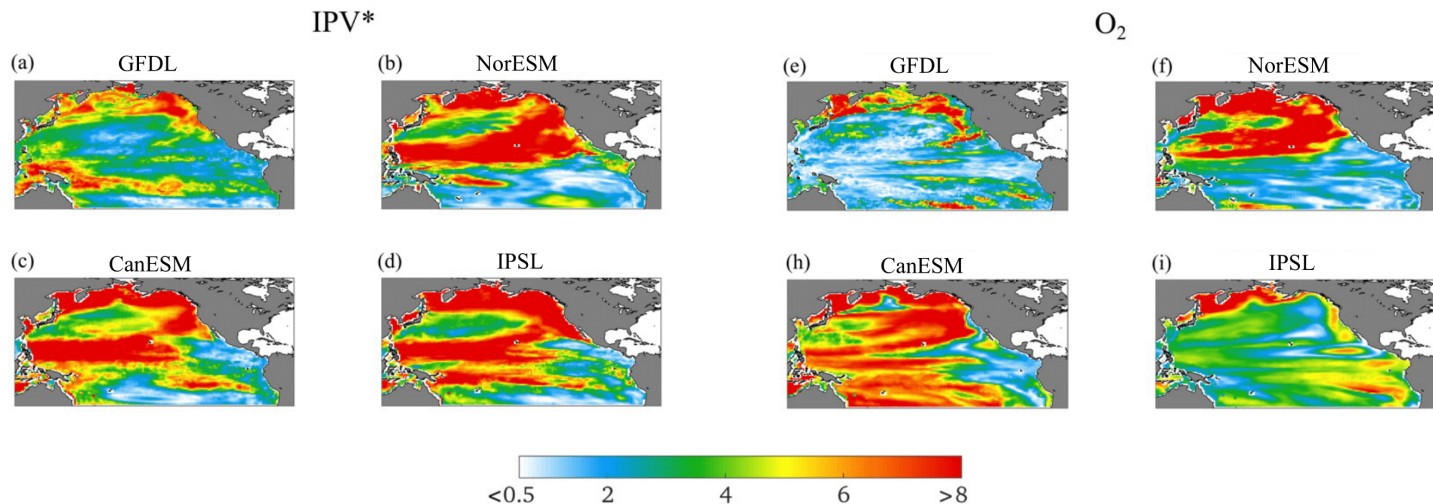

**Figure 13.** 2036-2100 SED *index* for IPV* (left) and $O_2$ (right). The colorscale is produced with rgbmap (Greene, 2023).

## 5. Discussion and Conclusions

State-of-the-art Earth System Models (ESMs) can simulate many aspects of the observed Earth's climate and biogeochemical processes, offering valuable insights into the future. Challenges persist, however, in representing reliably ocean biogeochemical dynamics (Schartau et al., 2017; Fennel et al., 2022). Biogeochemical processes can involve intricate interactions between multiple components of the earth system (Pascal et al., 2024). These processes are often nonlinear and coupling with physical climate processes are complex and challenging to interpret (e.g. Béal et al., 2010), therefore requiring

advances in diagnosis methods and interpretation. To assess model performance, continued efforts to develop metrics for model evaluation are needed. In this study we presented new tools stemming from data-mining techniques that may contribute to this end. These quantitative approaches, together with advances in observation-based gridded products, can better characterize and extract information about linkages between physical and biogeochemical variables. In particular, the availability of biogeochemical data, including dissolved oxygen, remains sparse compared to that of physical data. Limited observational data hinders model validation. Exploiting linkages between physical climate and oceanic $O_2$ can enhance understanding and predictive skills for biogeochemical tracers. Examples of recently developed tools that take advantage of these linkages can be found in Giglio et al. (2018) and Sharp et al. (2022), who applied machine learning tools to the ARGO-$O_2$ dataset to generate time-evolving maps of dissolved $O_2$ concentrations from seasonal to interannual timescales.

The overarching hypothesis in this work was that in the North Pacific the spatial-temporal variability of $O_2$ reflects that of ocean ventilation (Talley et al., 2011), which can be measured by the magnitude of the isopycnic potential vorticity (IPV). A recent study (Ito et al., 2019) found that at subtropical latitudes the variability of winter-time mixed layer depths and the subduction of $O_2$ are linked to the PDO. Elevated $O_2$ levels emerge downstream of the deepened winter mixed layer during the positive phase of the PDO. According to the same study, in the equatorial Pacific, the variability of upper ocean $O_2$ is linked to the stratification and the depth of thermocline, which in turn are modulated by the PDO. There has been a wide range of mechanisms suggested for the connection between upper ocean $O_2$ and ventilation, many of which can be represented in ESMs. We should note, however, that Ito et al., (2019) also showed that extra-tropical $O_2$ variability involves multiple types of physical-biogeochemical coupling that may compensate one another. For example, ventilation variability (Ridder & England, 2014; Duteil et al., 2014; McKinley et al., 2003) can have opposite imprint on $O_2$ than water mass shifts depending on the vertical stratification of temperature and $O_2$. In the subtropical thermocline, both temperature and $O_2$ decrease with depth, and vertical shifts of water masses generate positive correlation between them (Brandt et al., 2015; Duteil et al., 2014; Eddebbar et al., 2019). However, a negative relationship is expected between temperature and $O_2$ under ventilation-driven variability, as colder conditions are typically associated with stronger ventilation (thus higher $O_2$). The superposition of these two processes may cause partial compensations and could amplify inter-model differences, especially in $O_2$.

In this work tested the overarching hypothesis that the $O_2$ variability in the North Pacific is linked to that of ocean ventilation measured by the magnitude of the isopycnic potential vorticity, using four ESMs, a hindcast and reanalysis data. We verified the simplistic view that the spatial-temporal variability of $O_2$ reflects that of ocean ventilation through the analysis of potential predictability, of the linkages between ventilation and $O_2$ with the dominant climate modes of the North Pacific, and of the patterns of extreme events in ventilation and $O_2$. As tracer of physical ventilation, we chose Isopycnic Potential Vorticity or IPV*: a strong ventilation is assumed to generate a negative anomaly in IPV*, which then is advected and mixed by physical circulation and mixing processes. Ventilation supplies $O_2$-rich surface waters into the interior ocean, implying a

negative correlation between $O_2$ and IPV* as weak stratification (low IPV) may be linked to high oxygen. First, the information entropy (IE) was adopted to identify the areas where IPV* has a high predictability potential. Predictability was

generally high along two stripes enclosing the ENSO pattern and excluding the upwelling cold tongue regions, which were found to correspond to areas where $O_2$ and IPV* are strongly anticorrelated. The underlying mechanisms are relatively well understood (Ito et al., 2019) and this behavior is robust across all the analyzed datasets and does not change significantly in the future projections in the four ESMs. Therefore, around the Pacific Equator, IPV*, which is easily retrievable from temperature and salinity data, has a good predictability potential (higher than in the rest of the basin) and can be used as

proxy for $O_2$. The greater availability of temperature and salinity (and therefore stratification) observations from ARGO floats, reanalyses and modeled fields could be used in conjunction to the fewer co-located observations of $O_2$ to validate our findings and further extrapolate information about $O_2$ and its time evolution in these tropical areas.

Secondly, the variability of $O_2$ and IPV* was examined in relation to large-scale modes of climate variability in the extra-

tropical North Pacific. At mid-latitudes, the regional climate variability is PDO-dominated and our analysis shows very low predictability of IPV*, unlike the ENSO-dominated equatorial regions. The low predictability extends to the western boundary current and the Kuroshio-Oyashio extension region. In the extra-tropical North Pacific, the (linear) contribution of the PDO on $O_2$ and IPV*, and the trends of their residuals have comparable amplitude over the historical period. This is not verified in the future projections, when the trends become increasingly dominant. Pattern correlations in the PDO regression

maps (b coefficients) are generally quite small across models.

Thirdly, we evaluated the hotspots of change in IPV* and $O_2$ in the historical period and in the future projections. Overall, the historical hotspot indices or SED, computed separately for IPV* and $O_2$, suggest a longitudinal decoupling across the two variables for all datasets but for one model, NorESM. In addition, most of hotspots are in the extratropics. $O_2$ SED tend to be stronger along the eastern portion of the basin, while IPV* hotspots are mostly found over the western side of the basin and

along the southern boundary of the subtropical gyre. The intensity of the SED increases over time from the historical period to the end of the 21$^{st}$ century. Larger changes and hotspots are found at the gyre boundaries and in the northern portion of the basin, from the Kamchatka peninsula to the Gulf of Alaska. While $O_2$ loss is broadly linked to the strong increase in stratification, there are significant differences across model patterns, pointing to the need of further investigation. The existing uncertainty in the CMIP6 models' representation of oxygen changes limits the information that can be extracted

from the projections. We carried out our analysis only on a subsample of the CMIP6 catalog, but adding more models will not challenge this important conclusion. For a detailed model intercomparison of ocean deoxygenation in CMIP6 models the reader is referred to Abe and Minobe (2023). Major sources of uncertainty in the future projections reside, for example, in their ENSO-amplitude representation as detailed in Beobide-Arsuaga et al. (2021), and in uncertainties in the amount of future warming (Tokarska et al., 2020), and consequently in changes in upper-ocean stratification. Compared to the CMIP5

catalog, CMIP6 models tend to warm more, and show a decline of subsurface oxygen ventilation with no consistent decrease of inter-model uncertainties (Kwiatkowski et al. 2020). Here we found that while in some models the relationship between IPV* and $O_2$ becomes stronger, that is not the case for all, and it is not verified in GFDL, which has the highest horizontal resolution and best compares to the reanalyses in the historical period.

       A note of caution should be spent on the representation of regional changes and hotspots. The currently available spatial
resolution for CMIP6 models does not resolve the fine-scale (mesoscale and finer) physical and biogeochemical processes occurring near the coast. This is especially true in regions of elevated nutrient supply such as along the California Current System and more generally the Eastern Boundary Upwelling Systems (EBUS). Consequently, projected oxygen trends may exhibit variability even within subregions under the same scenario as shown, for EBUS, by Bograd et al. (2023). Analysis at the scales required to capture coastal dynamics, however, would require higher resolution models that will need – if
projected into the future – boundary conditions from CMIP6 simulations. CMIP6 models indeed remain the primary tool for evaluating changes in large-scale modes of climate variability at interannual to decadal times. While resolution is an important limitation for coastal areas, our main findings remain relevant in the interpretation of the large-scale forcing. In particular, the outcome of the hotspots analysis, i.e., that there is a large-scale longitudinal de-coupling between the areas of most prominent changes in IPV* and $O_2$ despite the PDO imprinting, is unlikely to be influenced by the models' resolution.

We found that the linkages between extra-tropical $O_2$ and PDO are model-dependent and the relationship not as strong as hypothesized on the base of the available sparse observations. In summary, the variability across the current generation of CMIP models is, for some of our hypotheses, too large to reach any definite conclusion on a signal which is weaker than expected. To alleviate this problem, we suggest using the new BGC-Argo array to validate the performance of each model by testing relationships between temperature, IPV and $O_2$.

Models and reanalyses or hindcasts such as E3SM-2G allow for testing if there may be predictability notwithstanding their biases, and for the case of the North Pacific, if there is a robust relationship across models between large-scale climate modes of variability, stratification and $O_2$. The predictability potential extrapolated by global ESMs represents an upper bound of the actual one, but it is useful for identifying when further exploration may be warranted or where such exercise may simply be futile. For example, the information entropy could be evaluated using ARGO data opportunely interpolated
(e.g., Smith and Murphy 2007, Cheng and Zhu 2016, and for BGC-ARGO Turner et al. 2023, Keppler et al. 2023, Sharp et al. 2022). In regions where the predictability potential is high, such an exercise is warranted, wherever the potential predictability is low, futile. In reference to our second hypothesis, we found that the PDO modulates IPV* and $O_2$, but the signal is not robust across models, limiting the possibility to reconstruct the large-scale evolution of $O_2$ from temperature and

salinity data alone. On the other hand, in the equatorial regions, generally under-sampled in historical $O_2$ datasets, the

relatively high predictability of IPV* and its strong link with $O_2$, could be exploited.

In summary, in this work we examined the relationship between the upper ocean (0-200 m) oxygen ($O_2$) content and stratification in the North Pacific Ocean in four CMIP6 ESMs, an ocean hindcast simulation, and an ocean reanalysis. As far as the robustness of $O_2$ and IPV* relation in the North Pacific (our first question), we found significant inter-model differences in the representation of climate variability in the North Pacific in CMIP6 models.

In relation to the linkages between $O_2$ and IPV* versus large-scale modes of climate variability such as PDO and ENSO (second question), we highlighted the potential of monitoring IPV* to infer $O_2$ evolution in the ENSO area. However, we did not find a robust signal in terms of patterns and time evolution in the extra-tropics, where the PDO is the dominant mode of climate variability.  The caveat is that the relationship, while weak, was nonetheless statistically significant under several metrics in the hindcast and in some models, GFDL-ESM4 being the best example.

Lastly, we found that the hotspots of changes in IPV* and $O_2$ are not co-located (third and final question), especially in the historical period.

In conclusion, the evolution trajectory of both stratification and oxygen in the North Pacific remains uncertain. Reducing this uncertainty would require monitoring simultaneously IPV and $O_2$, for example through the accumulation of ARGO floats

equipped with CTD and $O_2$ sensor, to better quantify the large-scale co-variability of physics and biogeochemistry as first step towards model improvement.

**Data and codes availability**

The python version of δ-MAPS is available at https://github.com/FabriFalasca/py-dMaps . The code for the Information Entropy computation is available at https://github.com/FabriFalasca/NonLinear_TimeSeries_Analysis . Climate indices used

in this study are from NOAA at https://psl.noaa.gov/data/climateindices/list/).  The CMIP6, Earth system model output used in this study is available via the Earth System Grid Federation (https://esgf-node.llnl.gov/search/cmip6/). The hotspots analysis was carried out using CDO (Schulzweida, U.: CDO User Guide (2.1.0). Zenodo. https://doi.org/10.5281/zenodo.7112925, 2022). A sample code for the hotspots calculation is also available at https://github.com/superlju/IPVO2hotspots/. We used the *eof* Python package (Dawson, 2016) for EOF analysis of spatial-

temporal data (available at https://ajdawson.github.io/eofs/latest/api/eofs.standard.html).

**Authors contribution**

LN performed all analysis, AB and TI conceived the study, TI and YT helped supervise the project, YT led the E3SM-2G development and integration. AB took the lead in writing the manuscript. All authors provided critical feedback and helped shape the research, analysis and manuscript.


**Competing interests**

The authors declare that they have no conflict of interest.

**Acknowledgements**

We thank Fabrizio Falasca and Ilias Fountalis, who developed several of the software tools for the data mining component of

this project. We acknowledge the Gibbs SeaWater (GSW) Oceanographic Toolbox for Python (https://teos-10.github.io/GSW-Python/intro.html ; IOC, SCOR and IAPSO, 2010; McDougall and Barker, 2011). The authors were supported by the Department of Energy, Regional and Global Model Analysis (RGMA) Program, Grant No. 0000253789. YT was supported by the U.S. Department of Energy, Office of Science, Office of Biological and Environmental Research, as part of the Energy Exascale Earth System Model project, and through NERC-NSF grant (C-STREAMS, reference

NE/W009579/1).

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
