# Peer review of "Evolution of oxygen and stratification in the North Pacific Ocean in CMIP6 Earth System Models"

_Biogeosciences, 2023_

## Author Comment (AC1)

Answers to referees for the submitted manuscript
**Evolution of oxygen and stratification in the North Pacific Ocean in CMIP6 Earth System Models**
L. Novi, A. Bracco, T. Ito and Y. Takano

**Referee #1**

Summary:

While the topic is certainly relevant, the approach used is novel and the motivation of the study is clearly justified, in my opinion this manuscript is only marginally suitable for Biogeosciences as its focus is mostly on the physical processes responsible for the supply of oxygen to the upper ocean. I therefore think that this manuscript is much more suitable for Ocean Science (also a Copernicus journal; see subject areas there). This does not however diminish the merit of this manuscript which upon moderate revision should be apt for publication in the appropriate journal.

While we understand the reviewer's concerns, we respectfully disagree. The topic, oxygen reconstruction, is of interest to ocean biologists far more than to physical oceanographers, and the readership of Biogeosciences represents the one we wish to reach. Using the words of reviewer 2: "The application to O2 is highly relevant as observations of oxygen are becoming more autonomous and is interesting on both physical and biogeochemical fronts because of its relationship to ventilation processes as well as ecosystem health."

General comments

Strengths:
The manuscript is mostly well written and the presentation of the results is of high quality. Except for a few areas that need clarification (see specific comments below), the manuscript provides an accurate account of the approaches followed by the authors to carry out their analyses. Generally speaking, the text is well structured and the connection between objectives, approach, results and the corresponding conclusions is clear and easy to follow.

We thank the reviewer for the overall positive assessment of our work.

Weaknesses:
- "After reading the manuscript it is not completely clear what is the level of uncertainty associated with the projections carried out with the different models (described as "future" in the text and figures). This should be clearly stated as it might impact the applicability of the suggested approach if it were to be employed (as suggested by the authors) with large-scale data from autonomous platforms."

  We state in the revised manuscript that the existing uncertainty in CMIP6 models' oxygen inventory changes poses a limit to the information that may be extracted in the future. While in some models the relationship between IPV* and O2 becomes stronger, that is not the case for all of them, and in particular for the GFDL model which has a better representation of the historical period than the others. We additionally point to the recent work of Abe and Minobe (2023) for a detailed model intercomparison of ocean deoxygenation in CMIP6 models, which was beyond the scope of our work. Additionally, we now better clarify that a major objective of this study is the presentation of a novel integrated approach to assess some of the major relationships between climate dynamics, upper ocean stratification and oxygen inventory, and to show if/when these relationships are robust across some CMIP6 models, despite their inherent uncertainties. We also add additional literature to better frame the limitations of our results (besides Abe and Minobe

(2023), we include Tokarska et al (2020), Beobide-Arsuaga et al (2021) and Kwiatkowski et al (2020).

- "While the study covers open and coastal ocean areas, from the plots it is evident that there is a limitation on the resolution in close-coastal areas. I think it would be worth stating the proximity to the coasts at which these analyses are valid, as biogeochemical processes affecting the oxygen distribution in e.g. eastern boundary upwelling systems are particularly intense within 50-150 km from the coast (do we expect the variability in near-coastal areas to be smoothed out in climatological time scales?)"

  Our work covers open ocean and coastal areas but the currently available resolution for CMIP6 models (or the publicly available observations for most part) does not allow for a detailed representation of the fine-scale (mesoscale and finer) physical and biogeochemical processes occurring near the coast except for specific coastal systems (for example the California Current System). While we acknowledge that this is a limitation, and an important one in the description of, for example, eastern boundaries upwelling systems (EBUS) biogeochemistry, our conclusions focus on large scale processes occurring in extended areas of the open ocean rather than on the coastal zones alone. In the North Pacific observations and models both suggest that there may be a large-scale modulation of oxygen content that extends to the coast. In addition, and most importantly, the main findings for the hotspots analysis, such as the large scale longitudinal de-coupling of hotspots and the extratropical location of changes, are unlikely to be influenced by the models' resolution approaching the coast.
  In the revision, we acknowledge that O2 concentrations in coastal areas are driven by complex biophysical interactions and physical processes unresolved by state-of-the-art climate models. Consequently, projected oxygen trends may exhibit variability even within subregions under the same scenario (see, again for EBUS, Bograd et al. (2023)). Analysis at the scales required to capture coastal dynamics, however, would require higher resolution models that will use - if projected into the future – CMIP6 runs as boundary conditions. Our work would remain relevant in the interpretation of the large scale forcing, while the overall methodology could be applied broadly even in higher resolution models/observations. We think it is important to clarify these aspects in the manuscript, and we better explain both limitations and applicability of our work in the revised version. We thank the reviewer for highlighting this point.

- "The results and conclusions section are rather puzzling. While the results in all subsections are well described, the discussion is limited and at the end one does not grasp the main message until reading the whole manuscript. My impression is that most of the current text on the conclusions actually corresponds to the missing text on the discussion. I recommend the authors to revise these two sections and cut down the conclusions to a brief text in which they state whether the study's goals (as laid out in the introduction) were achieved or not and why."

  We thank the reviewer for this comment, which we comply with by restructuring the discussion and conclusion sections of the manuscript as suggested.

- Specific comments (All taken care of)
l.122: replace "for which" by "in which".
l.143: replace "its" by "theirs"
l.235: I strongly suggest changing this section to "Results and Discussion".
l.243–244: Please elaborate on the criteria used to classify an area as "most impacted by ENSO".

l.252: "(…) where the variability is dominated by PDO (…)". Same comment as above for ENSO.
l.337–340: This sentence is long and not understandable at all. Please revise.
l.338: Remove apostrophe after "Peru".
l.375–378: This sentence is long and not understandable at all. Please revise.
l.426–437: This can be omitted or significantly reduced as this information is mostly redundant.
l.474: Consider changing to: "pointing to an area of further investigation".

We made all the requested minor corrections. We also clarify that we identified the ENSO and PDO impacted areas using the network inference and that we referred to the δ-MAPS domains in Figures 4-5.

- General comment to tables: according to Copernicus guidelines, "horizontal lines should normally only appear above and below the table, and as a separator between the head and the main body of the table". Please revise.

  We changed the table as suggested. Thanks for noticing it.

**References used in the answer to Referee#1:**

Abe, Y. and Minobe, S.: Comparison of ocean deoxygenation between CMIP models and an observational dataset in the North Pacific from 1958 to 2005. Frontiers in Marine Science. 10. 10.3389/fmars.2023.1161451, 2023

Beobide-Arsuaga, G., Bayr, T., Reintges, A., and Latif, M.: Uncertainty of ENSO-amplitude projections in CMIP5 and CMIP6 models, Clim. Dynam., 56, 3875–3888, https://doi.org/10.1007/s00382-021-05673-4, 2021.

Bograd, S. J. et al.: Climate change impacts on eastern boundary upwelling systems. *Annu. Rev. Mar. Sci.* **15**, 303–328, 2023.

Kwiatkowski, L., Torres, O., Bopp, L., Aumont, O., Chamberlain, M., Christian, J. R., Dunne, J. P., Gehlen, M., Ilyina, T., John, J. G., Lenton, A., Li, H., Lovenduski, N. S., Orr, J. C., Palmieri, J., Santana-Falcón, Y., Schwinger, J., Séférian, R., Stock, C. A., Tagliabue, A., Takano, Y., Tjiputra, J., Toyama, K., Tsujino, H., Watanabe, M., Yamamoto, A., Yool, A., and Ziehn, T.: Twenty-first century ocean warming, acidification, deoxygenation, and upper-ocean nutrient and primary production decline from CMIP6 model projections, Biogeosciences, 17, 3439–3470, https://doi.org/10.5194/bg-17-3439-2020, 2020

Tokarska, K. B., Stolpe, M. B., Sippel, S., Fischer, E. M., Smith, C. J., Lehner, F., and Knutti, R.: Past warming trend constrains future warming in CMIP6 models, Sci. Adv., 6, eaaz9549, https://doi.org/10.1126/sciadv.aaz9549, 2020.

---

## Author Comment (AC2)

**Evolution of oxygen and stratification in the North Pacific Ocean in CMIP6 Earth System Models**
L. Novi, A. Bracco, T. Ito and Y. Takano

**Referee #2**
The manuscript would benefit from much more explicit signposting throughout.
The analyses are impressive but are presented in such a way that they feel separate and the reader must make their own logical steps. In addition, the manuscript lacks a critical analysis of the results and their connection to existing literature on BGC reconstructions. Adding text to connect all the sections together and references to other literature will allow readers to better understand the method and its applicability to the problem of sparse observations of BGC variables, even outside of ocean oxygen.

We thank the reviewer for this general comment. We modified the presentation of the results following the above suggestions.

General Comments:

- "The manuscript would benefit from explicit signposting throughout its sections. The authors set out their hypotheses in the introduction but there are jumps between the results sections. The hypotheses focus on the regulation of O2 variability by IPV* yet most of the analyses approach IPV* and O2 separately (albeit connected by the PDO index)."

We added signposting explicitly linking each subsection of the results to one of hypotheses and we significantly restructured the main text to help clarify our message.

- "Additionally, hypothesis (3) seems to relate to regions where predictability is high, but in the results the identification of hotspots uses the residuals from the PDO regression, which is where predictability is low. The lack of signposting makes it difficult for the reader to understand the full implications of the previous analyses when a new analysis is introduced, and there is ambiguity as to how well the hypotheses have been answered in the conclusions section."

In the revision, we clarify that we isolate the PDO signal to verify if the low predictability in the N. Pacific (north of the ENSO region) is an issue of time scales (i.e. there is a low frequency PDO modulation with a high frequency 'noise' due to both atmospheric and oceanic variability. The PDO is indeed a lower frequency mode compared to ENSO and has most loading at higher latitudes, where weather 'noise' is greater). However, even when the PDO is isolated we do not find a strong anticorrelation between PDO-induced changes in IPV and PDO-induced changes in O2 in the E3SM forced simulation and there is large intermodel difference in the CMPI6 runs.
In the original manuscript, the rational for running the hotspot analysis on the residual fields was motivated by the fact that the PDO-forced component is low frequency variability and does not vary much over time (b, i.e. the regression coefficient between the PDO-forced physical fields and the PDO index, does not depend on time, which is only contained in the PDO index, which is shown). In addition, we had verified that the changes in the mean essentially coincide with the trends (therefore we were not removing any significant information). Taken together, these two considerations imply that the extremes will not depend on the PDO forced contribution. This is indeed the case, but we should have used the whole signal. We do so now, and both the maps of the indicators and SED are essentially unchanged (see for example the figures below for historical and future SED). We thank the referee for this comment.

[Figure]

Fig.1 Historical (and 1960-2014 period for the hindcast and ORAS4) SED index for the whole fields.

[Figure]

Fig.2 Future SED index for the whole fields.

- "I am not sure why the PDO was chosen as the potential proxy for understanding upper-ocean O2 concentrations. A priori I would assume that ENSO would play a significant role, and the results show a connection between ENSO and predictability of IPV* and O2, whereas when looking at the PDO alone there appears to be a limited connection. In the introduction Ito et al (2019) is mentioned but some further explanation would be helpful."

Our focus is on the North Pacific, as stated in the Introduction, and outside the tropical band the PDO modulates the variability at climate scales more so than ENSO. As also explored in previous literature (for example, Ito et al. (2019)), the dominant mode of oxygen variability in the northern Pacific Ocean is correlated with the PDO index which explains about 25% of the variance. To further verify it in the present work, we computed the first EOF for the E3SM-2G hindcast 0-200m O2 and IPV*anomalies over 1960-2014 over the northern Pacific (20.5°N-69.5°N;115.5°E-60.5°W) and the corresponding time series for the first principal component (*pc1*). The first EOF explains 25% of the oxygen variance and about 12% of the IPV*variance. The *–pc1* shows a significant and strong correlation (Pearson's R coefficient) with PDO timeseries computed using SST anomalies as detailed in the manuscript (R = 0.83, pval < 0.01 for O2, and R = -0.44, pval <0.01 for ipv*, after applying a 5-year moving means). The correlation with the PDO is higher than the one with ENSO, which is at most R = 0.34, pval < 0.01 for O2, after applying a 3-month moving mean. This is consistent with previous knowledge that a coherent basin-wide pattern of oxygen variability is mostly associated with PDO in the northern Pacific Ocean. We included in the revised *Introduction* a clarification of this

point and we discuss the outcome of the EOF analysis when indicating how we separate the PDO forced-component.

- "Additionally, I would be interested in seeing the regression analysis using both ENSO and the PDO as predictors and seeing how the residuals depend on the climate indices used for the regression."
We computed the linear regression analysis using the ENSO index, with the same procedure used for PDO, obtaining similar b shapes of the coefficients -as to be expected - but much lower absolute values, further confirming that the effect of PDO is overall dominant in the regression. Color limits are +/-3 standard deviations of the ensembles as in the main text. We included this finding in the Supplementary Material.

[Figure]

Fig. 3:

Fig. 3 $b_{IPV*}$ (left) and $b_{O2}$ (right) ENSO-regression coefficient maps with superposed contours of the ENSO and of the PDO+ and PDO- domains. The correlation coefficients among the corresponding maps for the same model or hindcast are also indicated. Color limits are fixed as +/- 3 standard deviations of the ensemble for each variable over the whole area.

- "There is no discussion section, and the conclusions section seems to repeat the findings of the study without any connection to the wider literature. I would recommend changing Section 5 to "Discussion and Conclusions" (or adding a separate Discussions section) and including a critical analysis on the relationship of this study to other work on the PDO and North Pacific oxygen (e.g., Ito et al 2019) or to reconstruction efforts (e.g. Sharp et al., 2022)."

We thank the referee for this comment. In the revision we provide a "Discussion and Conclusion" section that addresses the requested points. We extensively restructured the text to improve and extend it as suggested. Few examples of what we included:
We added a discussion on how our findings align with Ito et al (2019) in identifying a cohesive basin-wide prevailing pattern of oxygen variability in the northern Pacific primarily associated to the Pacific Decadal Oscillation.
We also discussed that reconstruction efforts (such as the one proposed in Sharp et al. (2022) for example) need to interpolate on a regular distribution, leading to additional uncertainty.
We stress that our framework could help determine the large-scale predictability potential of any field of interest.
We also included a discussion of the limitations of our work, as also stated elsewhere in this revision file. These include, for example, recognizing that oxygen

concentrations in coastal areas are influenced by complex biophysical interactions and physical processes that state-of-the-art climate models currently cannot fully resolve. As a result, projected oxygen trends may exhibit variability even within subregions under the same scenario, as demonstrated in studies like Bograd et al. (2023) for EBUS. Analyzing coastal dynamics at the required scales would necessitate higher resolution models, which, if projected into the future, would use, however, CMIP6 runs as boundary conditions.

A more extensive discussion and critical analysis with regard of existing literature is also provided in the revised manuscript.

- "I would like to see more discussion on the use of the CMIP6 ESMs. What are the implications of using relatively coarse-resolution models in this analysis? Particularly in the higher latitudes where eddy mixing is parameterized. What are the implications of having such a broad inter-model range in the results? It is not clear to me whether the emergent relationships are consistent enough across the models to justify the emphasis between IPV*, O2, and the PDO."

CMIP6 models are state-of-the-art in terms of climate prediction capabilities, but they are imperfect and limited in resolution. However, they are the only tool we have that allow for an evaluation of the decadal modes of climate variability, their impacts and their potential changes in a warming planet. Recognizing the biases these models have, we also analyzed a hindcast (E3SM-2G). Our goal is precisely to see if there is a relationship that is robust across models between large-scale climate modes of variability in the N Pacific and their impact on IPV and $O_2$. If this was the case, independently of the PDO or ENSO representation - which may differ in each model - we could reasonably conclude that IPV, which can be monitored, for example through ARGO floats, could be used to track the large-scale variability of $O_2$. This could also be done through models, where the resolution can be more easily increased if a simple (or no) biogeochemical module is included, and for which a validation can be conducted with less uncertainty on physical (instead of biogeochemical) variables due to the greater abundance of observations.

Indeed, the variability across the models is, for some of our hypotheses, too large to reach any conclusion and the relationship not as strong as hypothesized on the base of the available sparse observations. At the same time, at least for the historical period, our analysis allows for identifying which models may be more realistic in its representation of the large-scale variability of the North Pacific.

We expanded the Discussion and Conclusion section to include the points above.

- "I would like to see some discussion on the use of predictability studies for real-world reconstructions. As I understand it, the predictability mentioned in this study assumes perfect knowledge at a time *t* of a specific field, either IPV* or O2 (for Section 4.1) or sea surface temperatures (for the PDO index used in the regression analysis in Section 4.2). However, T and S profiles from Argo are still irregularly distributed, which is a nontrivial problem for ocean reconstructions of both heat and salinity themselves (e.g., Smith and Murphy 2007, Cheng and Zhu 2016) and ocean BGC (e.g., Turner et al. 2023, Keppler et al. 2023)".

Models and analyses or hindcasts allow for testing if a system is predictable - notwithstanding their biases -, or to calculate the potential predictability of a system. When using observations such as ARGO profiles, it is necessary to interpolate onto a regular distribution, which will increase the uncertainty. If the predictability potential is high, such an exercise will be worth it, if the potential predictability is low, futile. This assumes that the model(s) are capturing the main features driving the -in our case

large scale – predictability. We added a discussion point on the final section of the manuscript to reflect this comment.

Specific Comments:

Line 174: I am not familiar with this definition of extremes. Is there a reason you have not used a general quantile threshold or a distribution fit to characterize the extremes? As you use only one realization for each model, there is a nontrivial chance of "significant" changes in extremes due to internal variability.

We thank the referee for this comment. Building on previous works (Falasca et al. (2019); Falasca et al (2022)), we expect the topology of a given model to remain relatively stable, i.e. we do not expect the member choice to significantly influence the calculation of extremes and hotspots with the chosen definition. We verified the robustness of our results, computing the extremes indicators of four randomly-chosen ensemble members of the CanESM5 model for the whole signals of IPV* and O2 for the historical periods. We found no significant changes in extremes and SED, as shown in the figures below.

A major advantage of the hotspot definition chosen is that it accounts for changes in mean, variability and extremes at the same time in the identification of the hotspots. In other words, it accounts for the topology of the simulated climate fields, which can be characterized by considering all the three aspects together (as done also, implicitly, in δ-Maps).

The definition of extremes adopted aims at including information on seasons exceeding corresponding baseline extremes, without choosing a priori a threshold on the current distribution, which is especially relevant for comparing changes with respect to a reference baseline.

[Figure]

Fig4. O2 (left) and IPV*(right) indicators for changes in extremes (historical, whole signal) for four different ensemble members of the CanESM5 model.

[Figure]

Fig5. O2 (left) and IPV*(right) SED indices (historical, whole signal) for four different ensemble members of the CanESM5 model.

Line 189: Why is the depth horizon set to the top 200 m?
We set the depth horizon at 200m as it represents a reasonable trade-off between being in the upper thermocline, being deep enough to smooth the gas-exchange effects dominant at the surface, but not too deep as models tend to lose a good representation of variability (interannual and longer) when compared to observations.

Line 193: What is the reasoning behind the choice of ESM models for the ensemble? Without choosing multiple realizations for each model and using 4 models, the ensemble seems quite small relative to the available CMIP6 output. Also it would be good to know which biogeochemical models each ESM employs in Table 1, even if biological oxygen cycling is outside the scope of this manuscript.

The objective of this work is to present a set of metrics (a framework) that may help in identifying relationships and quantifying predictability potential across physical and biogeochemical variables.
With the goal above in mind, we chose a subsample of the CMIP6 catalog that did not resemble each other in terms of components and/or resolution. Adding more models will not challenge the main conclusions: 1) There remain significant intermodal differences in the representation of climate variability in the North Pacific. This is not just reflected in the patterns, but also in the representation of the relationships between physical (IPV) and biogeochemical (O2) variables, which is the focus of our investigation. 2) Such a relationship appears weaker than we hoped in all datasets analyzed, but is statistically significant under several metrics, in the hindcast and in some models (GFDL being the best example).
We clarify this point in the revised manuscript.

Line 206: JRA-55do v1.4 has an anticyclonic tropical cyclone in the NE Pacific in 1959 (as well as multiple anticyclonic tropical cyclones in the Atlantic, see https://climate.mri-jma.go.jp/pub/ocean/JRA55-do/). The issue is fixed in v1.5. It would be ideal to re-run the hindcast with the corrected atmospheric forcing. If that is not possible 1959 should be excluded from your analysis, perhaps using 1960-2014 as your historical period.

We thank the reviewer for the comment, as we were not aware of this problem. We re-ran all the computations involving E3SM-2G and ORAS4 (for consistency of comparison) over 1960-2014. Results are nearly identical, as one year alone does not modify the PDO or the overall Pacific variability (see figure below). In particular, for both E3SM-2G and ORAS4, we re-run the entropy regression, and hotspots analyses over the new period. For the latter,

we divided 1960-2014 in two intervals of equal length, 1960-1986 and 1988-2014. We replaced these new analyses in the revised manuscript.

[Figure]

Fig. 6: PDO (left) and ENSO (right) indices (SST cumulative anomalies) calculated using δ-MAPS in the historical period, using E3SM-2G over 1958-2014 (green) and 1960-2014 (black). The numbers on top-right of the panels are the correlation coefficients between the two curves where they overlap in time, i.e. 1960-2014.

Line 238: I am not sure exactly what predictability means. Based on (1), this assumes some perfect knowledge of t=0 everywhere in the North Pacific for each of these models? What is the length of time used to calculate the IE? Perhaps I misunderstand something in the methods with these questions, but clearer definitions for predictability both here and in the methods would be helpful.

We thank the referee for pointing out the need for additional clarifications. We added a more detailed explanation of predictability as linked to the entropy measure in the method section. The quantification of IE relies on recurrence. In each point, the entropy of the field under investigation is associated with recurring microstates in its time series (that define the system and thereby impacts its predictability). The higher the predictability of a time series the more recurrent are its temporal dynamics, i.e. the easiest will be to predict its future evolution. We computed IE using monthly data over the whole historical and future periods.

Line 244: What do you mean by "area most impacted by ENSO"? Has there been a regression analysis done for ENSO in each of the models? Figure 2 a-f seem to have quite high IE (low predictability) in the equatorial upwelling region across all the models.

We rephrased it to indicate more clearly that we mean. Here we meant the domain identified as ENSO-related by δ-Maps, which well follows what would be identified by an EOF analysis over the SST field as region where the variance explained by PC1 is greatest. As stated in the manuscript "The predictability potential is higher along two stripes enclosing the ENSO domain but excluding the upwelling cold tongue regions." This result is not new. The predictability of the cold tongue has been found to be low over much longer periods in the IPSL model (Falasca et al. (2020)), and in SODA reanalysis and a large suite of CMIP5 models in Ikuyajolu et al (2021).

Line 319-320: How do the regression coefficients stay relatively stable if the domain for the PDO evolves? Also, what is the implication about the residuals dominating the evolution of both IPV* and O2 in terms of predictability (and, in particular, predictability related to the PDO)?
The regression coefficients are computed point by point using the local value of O2 (or IPV*) timeseries, and the PDO time-series (the same for all grid points). The PDO signal is computed using the timeseries associated to the PDO domains, which capture the overall decadal climate variability regardless of their exact shape, and this decadal variability is not changing significantly in most cases (the PDO dynamics are not changing significantly, which is not surprising).

Technical Comments:

Line 119: Should xi be multidimensional?
Thank you for catching this typo. Yes, xi is indeed multi-dimensional. Also, in equation (1) of the revised version, we replaced xi∈ℝ with **xi**∈ℝ^d, being d the **xi** space dimension.

Line 135: What is the reasoning behind the use of 4 microstates?
We ran a sensitivity analysis of the entropy field to m, within a meaningful range according to previous literature (Ikuyajolu et al. (2021)). We tested m = 2,3,4,5 for GFDL-ESM4 over 1950-2014. In the figure below we show the results when the same color scale is used (panels a-d) and when each case has a different color scale to highlight the spatial features (panels e-h). The pattern, i.e. areas more (less) predictable relative to the surroundings are substantially unchanged, i.e. the geographical patterns are robust, as also found in Ikuyajolu et al. m = 4 and m = 5 show reasonable entropy values, but we chose m=4 is because it spans the widest range of possible values, as also shown by the histogram below.

[Figure]

Fig.7: Historical (1950-2014) GFDL-ESM4 Entropy maps computed using m=1,2,3,4. Panels (a)-(d) have the same color scale than the one used in the main text. Each panel from (e) to (h) (same fields as panels (a)-(d)) has a diffent color scale, to show the spatial pattern.

[Figure]

Fig.8: Histograms for historical (1950-2014) GFDL-ESM4 Entropy maps computed using m=1,2,3,4.

Line 160: Why have you not used the same years across the models and reanalysis and hindcast product for each period?
We are looking at a decadal variability mode, the PDO, as main focus of our study, and wanted to cover the longest possible period for the models and as further as possible in the future projections (historical, 1950-2014 and future 2036-2100), but the reanalysis and hindcast are only available over a shorter time period. Therefore, we decided to keep the model on a longer time range to capture the PDO temporal scales as best as possible.

Line 162: I find the use of shortcuts like *Period 1/2, Ind 1/2/3, yseasm* to hinder my understanding, particularly when examining the figures. More descriptive shortcuts (e.g., \overbar{DJF1983-2014} - \overbar{DJF1950-1981} instead of *Ind1* ) would greatly help readability
We changed notations in the revised manuscript as recommended.

Answers to all minor points is after the comments list:

Line 186: perhaps define N2 here?
Line 188: Do you mean Equation 4?
Line 195: Which variables are used from the CMIP6 models? If T and S, it would be helpful to know which models use EOS-80 and which use TEOS-10 for their density fields.
Line 196: Do you mean SSP?
Line 207: ORAS4 could use a description in this section. Also to explain about the lack of O2 results (I presume the reanalysis has no biogeochemistry?)
Line 221: RMSE values embedded within Figure 1 rather than presented as a list would increase readability of this section.
Line 223 and elsewhere: Please use consistent units formatting with superscripts.
Line 279: Is it possible to include a scaled version of the NOAA PDO time series in Figure 3 for comparison?
Line 294: Is there one bO2 and bIPV* for all scenarios or are the coefficients calculated for each scenario separately?
Line 335: ORAS4
Line 435: Repeat here the vertical domain (0-200m)
We thank the referee for catching some typos and giving recommendations for improvements. In the revised manuscript we implemented all the corrections and requested changes. We answer to the questions as follows:
L195: That is correct, the variables from the CMIP6 models are potential temperature and salinity. The requested information will be included in the revised table describing the models.
L207: Yes, that is correct. We will add a clarification in the revised text.
L294: The coefficients are computed for each dataset separately.

**References used in the answer to Referee#2**

Bograd, S. J. et al.: Climate change impacts on eastern boundary upwelling systems. *Annu. Rev. Mar. Sci*. **15**, 303–328, 2023.

Ikuyajolu, O. J., Falasca, F. and Bracco, A.: Information Entropy as Quantifier of Potential Predictability in the Tropical Indo-Pacific Basin. Front. Clim. 3:675840. doi: 10.3389/fclim.2021.675840, 2021.

Ito, T., Long, M. C., Deutsch, C., Minobe, S., and Sun, D.: Mechanisms of low-frequency oxygen variability in the North Pacific. Global Biogeochemical Cycles, 33(2), 110–124. https://doi.org/10.1029/2018GB005987, 2019

Falasca, F., Bracco, A., Nenes, A., and Fountalis, I.: Dimensionality Reduction and Network Inference for Climate Data Using $\delta$-MAPS: Application to the CESM Large Ensemble Sea Surface Temperature, J. Adv. Model. Earth Syst., 11, 1479–1515, 2019.

Falasca, F., Crétat, J., Braconnot, P., and Bracco, A.: Spatiotemporal complexity and time-dependent networks in sea surface temperature from mid- to late Holocene, Europ. Phys. J. Plus, 135, 1–21, https://doi.org/10.1140/epjp/s13360-020-00403-x, 2020

---

## Author Comment (AC4)

**Evolution of oxygen and stratification in the North Pacific Ocean in CMIP6 Earth System Models**
L. Novi, A. Bracco, T. Ito and Y. Takano

**Referee #3**
Clarifications to the methods for non-specialists

- I am not familiar with the methods regarding data-mining tools ($\partial$-Maps) nor Information Entropy (IE). I found the explanation of how IE was calculated very well put, and I was able to follow without much difficulty. The exception here, however, was when the authors state "the explicit dependence of the entropy quantifier one is removed using the maximum entropy formulation". At this point I was not sure of what the authors were doing, since the way an explicit e is removed is explained in *Ikuyajolu et al (2021)* that the authors point to and I am not familiar with.
  We added a sentence to explain this point. More in detail, the recent work of Prado et al. (2020) proposes a method to free the entropy calculation from the selection a distance threshold (epsilon). They analyzed the dependency of Entropy on epsilon, and found that the Entropy-epsilon relationship has a well-defined maximum (Smax in Fig 4 of Prado et al. (2020)), which is robust and relatively stable within a range of epsilon values, and that this maximum entropy is strongly correlated with the Lyapunov exponent. In our work, we applied the heuristic explained in detail in Ikuyajolu et al (2021) for the calculation of Smax through an iterative procedure that calculates the recurrence entropy for varying epsilon until a maximum is found and retained. This algorithm requires three input parameters: the microstate dimension (that we set at m=4 but explored other values as well in the revision), the number of random samples to compute the microstates distribution in the RP (here 10000) and the number of random sub-samples used to determine the epsilon for which entropy is max (here 1000).

- Another thing, am I interpreting things correctly if the choice to use 4 microstates means that their IE calculation uses 4 probabilities of occurrence (k=4 in equation 2)? Doesn't this mean that the authors are choosing four different e values to create these 4 microstates from the same timeseries of IPV* Euclidean Distances (Eqn. 1)? So here I am confused about how e and Eqn. 1 is actually being done.

  We thank the reviewer for having pointed this out, and we acknowledge that the explanation was not clear, therefore we corrected it in the revised version. In particular, the sentence "using 4 microstates" at line 134 of the initially submitted manuscript should have been instead "using microstates of dimension 4". Indeed, the dimension of a microstate is the size of the NxN matrix introduced at line 124 (of the initially submitted version), i.e. N=4 in this case. Microstates are calculated sampling matrices of size NxN inside the recurrence plot (RP), and the total number of possible configurations of a microstate of dimension N is $N_{tot} = 2^{(N^2)}$. Therefore, the Probability of occurrence of the generic $k^{th}$ microstate is $P_k$ is used in eq (2) of the initially submitted version and detailed at line 124 (initially submitted version). Therefore, that equation uses $N_{tot}$ different values in the summation. We thank very much the referee for this useful comment, as it helped clarify the method in the revised version.
  We also note that in response to Rev 2, we will be adding in the supplementary a sensitivity analysis based on the number of microstates.

- It is also not clear to me on reading the methods how you calculate the 95th percentile of mean, variability and extremes in Eqn. 3. For the mean (indicator 1), as an example, are you computing the differences across all years in Period 1, then calculating the 95th percentile of these differences? But actually, this doesn't seem to be the case, because you state that *ind1j* is equal to $yseasm2 − yseasm)$, *where yseasm1 and yseasm2 are multi-*

*year seasonal means in Period 1 and 2, respectively*. Multi-year is arbitrary, and on first reading I am inclined to believe that its averaging across the whole length of the period. This suggests to me that *ind1j* is one number, and so it is not clear to me how you then retrieve a 95th percentile. Could you please make this explanation clearer?

The indicators and SEDs are computed point by point, i.e. each grid point has one value. The percentile is therefore computed spatially over all the grid points. We added a sentence in the text to clarify this point.

Other recommendations:
I very much agree with Reviewer 2 in that the paper would benefit from more signposting throughout, and that a reader only really appreciates what they have learned from the results in the final sentences of the conclusions.
I also agree that there is probably additional studies to point to so that the work can be placed amongst the wider literature.
We have addressed all reviewer 2' comments and restructured greatly the presentation of the results to help the readers following the presentation of all the analyses and how they are relevant to the stated hypotheses.

"There is also very little discussion or mention of the other processes affecting O2. Apart from ventilation, there is also the solubility effect of warming and biogeochemical processes affecting oxygen demand. I think the paper would benefit from a discussion of how these two other factors come into play. For the solubility effect, it is not obvious how one would separate it from the ventilation component captured by IPV, since both are driven by warming and the IPV-O2 relationship should actually encompass both the ventilation and solubility effects. I leave it to the authors to think about how the solubility effect could be separated from the ventilation effect. However, for the biogeochemical processes, it would not be so difficult to calculate preformed O2 from the T and S of each model and redo some of your analysis. An analysis of the IPV* - preformed O2 relationship would eliminate any impact of the biogeochemical process, and then allow you to focus on predictability of physical O2 injection. I would expect substantially more predictability and over a much greater area. Similarly, you could take preformed O2 away from your O2 tracer to get AOU, and you could look at the predictability of AOU, which is likely not predictable at all from IPV? Maybe give this a try, and see if some interesting results jump out. This would at least allow you diagnose why IPV-O2 predictability falls over in some regions?"
We thank the referee for this insightful comment. We re-ran all calculations to evaluate the IPV*-linked predictability potential for AOU, i.e. the areas where IPV* and AOU time series are positively correlated with correlation coefficients > 0.5. These areas are very similar to the ones obtained by studying the relationship O2-IPV* proposed in the submitted manuscript. We also separated the solubility part O2sol, which is a very good approximation for preformed O2 at the considered depths, and computed the anticorrelations areas (i.e. where c.c. < -0.5, as for the IPV*-O2 relationship). Interestingly, these areas are not too small (especially in the hindcast) but are mostly superimposed to high-entropy/low predictability areas for IPV*. These results are reported in the figures below. We added a discussion of the two contributions in the revision and the difference in spatial patterns.

[Figure]

Fig.1: IPV* entropy field in the historical interval for the ESMs, and in the historical 1960-2014 period for the hindcast and ORAS4 with superposed the contours of the areas where IPV* and AOU time series are correlated with correlation coefficients > 0.5 (left), and where IPV* and $O_{2sol}$ time series are anticorrelated with correlation coefficients < -0.5 (right)

Specific comments:

- Line 182: More accurate to say "We obtain three indicators grouped into four seasons for each variable"?
  We thank the referee for the proposed rewording. This is correct and clearer that what stated, so we updated the text as suggested.

- Line 214: I know gridded Argo doesn't provide T and S as far back as your period 1, but couldn't you compare the ORAS4 against gridded Argo in Period 2? This would then provide some measure of how much bias there is in ORAS4, with which you are then using to assess bias in the models. Case in point is that the IPSL IPV fields looks (at least by eye) the least biased. It is not a coincidence because the IPSL and ORAS4 both use NEMO as their ocean models.

  Thank you for this comment, we attach below the comparison between the ORAS4 IPV* climatology over the "period 2" (1988-2014 for this reanalysis) with the corresponding climatology computed using SODA3.4.2, which is suitable as it uses a different ocean component compared to ORAS4 and overcomes the limitations of T and S data from ARGO before 2002. As shown in the figure below, the difference across reanalysis that use different models but assimilate the same observations is much smaller (about one order of magnitude) that the signal, and smaller than any model bias. We will include this information in the Supplementary Material.

[Figure]

Fig.2: Comparison between IPV* 1988-2014 climatology computed with ORAS4 and SODA.3.4.2.

- Line 221: How is it that the RMSE of the NorESM2-LM is the lowest among the models? Are you sure this is calculated correctly?
  Thank you for this comment. We verified the rmse calculation with three different scripts, two of them in Matlab and one of them via cdo ("climate data operators"). In cdo, for example, we used the following: cdo -L -sqrt -fldmean -sqr -sub model_nor.nc rean.nc rmse_nor.nc
  where model_nor.nc and rean.nc are the IPV* climatology over 1950-2014 (multiyear seasonal means computed with cdo) of NorESM2-LM and ORAS4 respectively.
  We always obtained the same result, $4.92 * 10^{-9}$. We also verified that the climatologies were computed correctly. The two extratropical areas where the bias is higher (which we agree make the visual estimation of rmse hard) are likely more than counter-balanced by the tropical areas where the bias is small. In the revised version, we updated both the plots and the RMSE calculations to comply with what requested by Referee#2, i.e. to re-do all the calculations involving E3SM-2G over 1960-2014 (instead of 1958-2014 for these members) with no significant differences.

- Line 307: Maybe remind the reader here was Ind1 is.
  Thank you, we added a sentence in the text. We also followed the advice of referee#2 are used a more readable notation throughout the revised manuscript.

- Figures 4 and 5: Add text to the legend stating a more intuitive way of interpreting the plots. For *bIPV\** you are presenting the change in IPV* (m-1 s-1) per change in SST (ºC-1), right?
  Yes, this is correct. We updated the figures as requested.

- Is the MPAS-O ocean model based on some version of MOM? The correspondence between the two models is striking.
  No, the MPAS-O ocean model is not based on MOM. See Ringler et al (2013).

Supplementary Material comments:

Figure S4: you've stated 1950-2014 twice?
Thank you for catching this typo. It is now corrected.

Thank you for considering my input to your research,
Pearse J. Buchanan.

**References used in the answer to Referee # 3**

Ikuyajolu, O. J., Falasca, F. and Bracco, A.: Information Entropy as Quantifier of Potential Predictability in the Tropical Indo-Pacific Basin. Front. Clim. 3:675840. doi: 10.3389/fclim.2021.675840, 2021.

Prado, T., Corso, G., Santos Lima, G., Budzinski, R., Boaretto, B., Ferrari, F., Macau, E.E.N. and Lopes, S.R.: Maximum entropy principle in recurrence plot analysis on stochastic and chaotic systems. Chaos 30:043123. doi: 10.1063/1.5125921, 2020

Ringler, T., Petersen, M., Higdon, R. L., Jacobsen, D., Jones, P. W., and Maltrud, M.: A multi-resolution approach to global ocean modeling, Ocean Model., 69, 211–232, doi:10.1016/j.ocemod.2013.04.010, 2013

---

## Referee Report (RR1)

Thanks to the authors for providing an updated manuscript. Many of the author's responses have clarified points for me, and after gaining a better understanding of the research I find I still have many questions about the work. Therefore I recommend that there be a second major revision for the manuscript. I hope that my points help make the manuscript more accessible to an interested audience, as I still believe there can be a lot of value in using IE for ocean biogeochemistry. I am a bit disappointed that some of my concerns have not been addressed in this new manuscript. I have repeated those concerns within my review.

**Major points:**

**Overall manuscript**
There still needs to be much serious consideration on how best to frame this work for its intended audience. I appreciate the consideration about the mathematics behind information entropy and the mapping procedure; however, *Biogeosciences* reaches a wide variety of scientists interested in the Earth system with varying backgrounds in mathematics. Some of the references, like those to topology, can be easily misinterpreted or misunderstood by readers unfamiliar with the mathematics. Even if the terms are correct they can present a barrier to the audience.

I wonder if the title can more clearly reflect how the work uses stratification as a proxy to understand ocean oxygen and its evolution. Currently the manuscript sounds like a model intercomparison which it is not.

There are numerous errors in language use, generally in the use of gerunds and matching up singular/plural nouns and verbs. It makes parts of the manuscript quite difficult to understand. I cannot provide every example but I would strongly recommend the authors conduct a thorough proofreading for the manuscript.

**Introduction**
There is no reference to current reconstruction schemes of ocean oxygen, or indeed the fact that observations of oxygen are sparse in space and time. As it is part of both the abstract and the discussion it should be included in the introduction.

I appreciate the highlighting of the hypotheses and the questions in this new version, but the paper should be framed such that it references, in order:
   1. The aim/objectives of the paper
   2. The questions the paper aims to answer
   3. The hypotheses for these questions
Right now the order is objective -> hypotheses -> questions, which makes it difficult to read. Additionally I do not think that the hypotheses need to be referenced as HYP1/2/3, as this just makes the manuscript more difficult to read and reference.

There is no literature review on the use of IE in oceanography. Any citations about its application exist much later and they should absolutely be included in the introduction to provide

justification for this work. In addition, there needs to be a section in which the use of IE for physical properties like temperature is compared to its potential use for ocean biogeochemical properties. If space is an issue, this is something that could replace parts of the beginning discussion on ocean oxygen cycling which is a bit high-level, has very few references, and also mentions quite a few biological processes which are ignored for the rest of the manuscript.

Lines 113-115 seem like an additional objective in the manuscript. If this is the case it needs to be reformatted.

Generally an introduction ends with an outline of the following sections. In the current introduction there is too much emphasis on the methods (which should be in the immediately-following section). A brief outline of what is included in the methods can be in the section introducing the following sections.

**Materials and methods**
Which historical experiments are being used? ESMs have emissions-based historical forcings and concentration-based historical forcings. In either case the radiative forcing is not necessarily the same between models with different atmospheric components and atmospheric chemistry.

It would help to include which variables you take from the CMIP6 models. Also, please define how you calculate density from temperature and salinity and how it is implemented in Equation 4. Is it using EOS80 or TEOS10? Are all model outputs gridded so that 0-200m are full cells, or is there some interpolation involved?

I still find the names of these indices (Ind1, Ind2) to be very difficult to internalize, both in the methods section and afterwards. I would strongly suggest eliminating the shorthand entirely. For the definitions on lines 191-193 I would strongly suggest using an overbar to indicate time averages rather than angle brackets.

I am still unsure about the arguments about the robustness of the extreme metric. If a 32 year time series of, say, winter stratification is used, the extreme is effectively the 97th percentile. This definition is necessarily dependent on both the length of the time series used and the number of ensemble members used. Additionally, is there a reason why you use seasonal averages to calculate the extremes, but then divide by the total number of months within that season? I am not sure if I am missing something major here but this continues to confuse me.

**Results**
The manuscript mentions that the hindcast "not surprisingly" displays the best fit for IPV and O2. This makes me wonder about the usefulness of a fully-coupled ESM compared to other ocean BGC models forced by reanalyses. What are the benefits of using a fully-coupled ESM? If the representation of the PDO is the largest issue in reproducing IPV and O2 (although I'm not convinced this is true) then I would assume this work would be more successful using ocean-only runs. Coupled models exhibit a "signal to noise paradox" (e.g., Scaife and Smith

2018, Zhang et al 2021) which results in an apparent decrease in predictability that may be restricted to model world. How would this impact the interpretability of your results?

I am a bit confused about the argument about how low-frequency PDO modulation alongside higher-frequency variability can produce low predictability. By that argument there is information associated with the PDO, but it is just more difficult to isolate. Is this not a case where the timescales of interest need to be controlled for? You use interannual changes in seasonal values for IPV and oxygen, but if we know the PDO to operate on decadal timescales, is there any way that the IE method can be altered to take these longer timescales into account?

Please ensure that all figures follow inclusivity guidelines. In particular Figures 4, 5, 9, and 13 need to be revised so that readers with deuteranomaly (red-green colourblindness) can interpret the figures.

Additional points on figure legibility:
- Figure 1: Please indicate which panels illustrate biases within the figure itself and ensure that "RMSE = " is included in each subplot.
- Figure 2: The years for the historical and future runs should be included in the figure, as well as some reference to IE so that the reader can differentiate the figure from later figures. Also, the colorbar direction seems counterintuitive to me, as I would normally associate green with "good" (i.e., lots of information) and this is actually for high entropy.
- Figure 3: Change the x-ticks for the historical run to green and the future scenario run to black. Subpanel b needs a similar label to differentiate the green and black dashed lines.
- Figures 4 & 5: the contours for ENSO and PDO are difficult to separate for some panels. Please use dashed lines for one of the indices so that the reader can more easily discern which regions are related to ENSO and PDO.
- Figure 6: The years in the caption are wrong and they should reflect the two periods used in the averaging rather than combine them into one period.

**Discussion/Conclusions**
Thanks for including a discussion. I think there still needs to be many more citations. For instance, the first page cites only Ito et al 2019. On line 534-535 you reference a wide range of mechanisms but there is no relevant literature included.

In the discussion, the manuscript references that reconstructions need to interpolate onto a regular grid. This is fleshed out more in the authors' response. This is not true – there is no inherent requirement to interpolate observations *or* innovations onto a regular grid when creating a reconstruction. There is utility in doing this, particularly as reconstructions are often used to compare or benchmark models, but any further assertion should be removed from the manuscript.

Please include a concluding paragraph that restates the objectives and main findings of the paper.

**Minor points:**

Line 20: "may or may not differ among the two fields, with a strong model dependency" what does this mean?

Line 39: Please include citations to OMZ literature.

Line 59-60: The reference to IPV is difficult to understand, as it seems to reference an equation but yet has no equation in the vicinity. I would recommend either moving the definition to the methods or having a more first-order reference to IPV here.

Line 63: It is not solely large-scale ocean currents that determines the pathways of IPV and $O_2$.

Line 97: "For example, given a time series,..." this example is unclear. What do you mean by information? Please explain in terms applicable to a general Earth Science audience.

Line 103-104: "It also allows to investigate the network of domains…" if this work is not done in the manuscript I would recommend deleting this.

Line 142: There is no need to reference a specific figure in another work.

Line 206: Please ensure all equations are given their own line and equation number.

Line 249: What sort of remapping is used?

Line 382: "as to be expected" Please refrain from using asides in the manuscript.

Line 411: It is not 2015 values that you are comparing but a longer averaging period, correct?

Line 515: I would say "with good accuracy" is not necessarily correct.

Line 613-614: Please rephrase the sentence about how IE can be connected to ocean reconstructions. I do not feel it is appropriate to imply futility in existing scientific research.

**References**
Scaife, A.A., Smith, D. A signal-to-noise paradox in climate science. *npj Clim Atmos Sci* **1**, 28 (2018). https://doi.org/10.1038/s41612-018-0038-4

Zhang, W., Kirtman, B., Siqueira, L. *et al.* Understanding the signal-to-noise paradox in decadal climate predictability from CMIP5 and an eddying global coupled model. *Clim Dyn* **56**, 2895–2913 (2021). https://doi.org/10.1007/s00382-020-05621-8

---

## Author Response (AR2)

Answers to referees for the submitted manuscript (revised title, as requested):

**Evolution of oxygen, stratification and their relationship in the North Pacific Ocean in CMIP6 Earth System Models**

L. Novi, A. Bracco, T. Ito and Y. Takano

Dear Editor,

Thank you very much for your consideration. We report below our point-by-point answers to the referees, which also detail the implemented changes.

**Referee#3 – Second review.**

"Summary
Novi et al have improved the manuscript in this revision."

We thank the referee for this positive evaluation.

"Clarifications to the methods for non-specialists:
I am pleased with the edits and additions that Novi & colleagues have made to explain their approaches. I also agree with them that these approaches are useful for oceanography and biogeochemistry, and I hope myself to learn and implement them in my own work. On this note, can the authors provide a link to their analysis code on their Github? If the authors are serious about uptake of these techniques by the community, I think that providing their "publication ready" code would go a long way to achieving this goal."

We thank the referee for this positive feedback and appreciation of the improvements in our work. The Data Availability section contains the codes that we used in this work. We changed the title of that section to "Data and codes availability" to point the reader to that section. In particular, that section contains the GitHub links to the python code for δ-MAPS and for the Information Entropy calculations. For the hotspots calculation, we used a collection of command line arguments via CDO-climate data operator, cited in the References as Schulzweida, U.: CDO User Guide (2.1.0). Zenodo. https://doi.org/10.5281/zenodo.7112925, 2022. We pointed to CDO also in the Data and codes availability section. Additionally, we made available a sample code for the hotspots calculation, now available at https://github.com/superlju/IPVO2hotspots/tree/main

"Signposting
The addition of hypotheses has improved the signposting."

We thank the referee for this positive feedback.

"Line 340: I think a more informative title would be "Climate indices and their relationship with the IPV and O2". Something like this or related."

We thank the referee for this recommendation. We changed the title to "Estimation of climate indices and their relationship with IPV* and O2".

Line 399: Can you end this section with something that tells the reader what we have learned?

We thank the referee for this suggestion. We have restructure and updated this paragraph in order to satisfy this request.

"Further analyses
Thanks for including the analysis on AOU and O2 solubility. While I was disappointed that this analysis wasn't carried through the paper into the predictability (hypothesis 2) and hotspot (hypothesis 3) sections, I understand that the paper is large enough as it is."

We thank the referee for appreciating our work. The discussion we included on the AOU and O2sol analysis (Lines 309-314 of the first revision and Suppl Fig S5) relates indeed to the predictability potential (paragraph 4.1). We acknowledge that we didn't include the AOU and O2sol in the hotspots analysis because – as also stated by the referee – we preferred to keep the manuscript length concise. We believe however that the analysis of hotspots of AOU and O2sol could be an interesting topic to address in a separate work.

Specific comments:
• Line 73: Replace "oceans" with "North Pacific".
Thank you, we have updated the sentence as requested.

• Line 543: What do you mean by "challenge the overarching hypothesis?"
The overarching hypothesis in this work was that in the North Pacific the spatial-temporal variability of $O_2$ reflects that of ocean ventilation, which can be measured by the magnitude of the isopycnic potential vorticity (IPV). By that sentence we meant to test the hypothesis stated above. We reframed the sentence in the manuscript to better clarify this point.

Thank you for considering my input to your research,
Pearse J. Buchanan.
* * *
**Referee#2 – Second review.**

Thanks to the authors for providing an updated manuscript. Many of the author's responses have clarified points for me, and after gaining a better understanding of the research I find I still have many questions about the work. Therefore I recommend that there be a second major revision for the manuscript. I hope that my points help make the manuscript more accessible to an interested audience, as I still believe there can be a lot of value in using IE for ocean biogeochemistry. I am a bit disappointed that some of my concerns have not been addressed in this new manuscript. I have repeated those concerns within my review.

**Major points:**

**Overall manuscript**

"There still needs to be much serious consideration on how best to frame this work for its intended audience. I appreciate the consideration about the mathematics behind information entropy and the mapping procedure; however, *Biogeosciences* reaches a wide variety of scientists interested in the Earth system with varying backgrounds in mathematics. Some of the references, like those to topology, can be easily misinterpreted or misunderstood by readers unfamiliar with the mathematics. Even if the terms are correct they can present a barrier to the audience."

We introduce methods not commonly used in the ocean biogeochemistry community, providing a novel framework to assess predictability or changes, focusing on oxygen and IPV. In our first submission, we limited the description of the methods to their generalities, but we were asked to provide more details. Referee#3 explicitly asked for "Clarifications to the methods for non-specialists" (see first report of Referee#3) with a particular focus on the IE and the SED calculation. In response to our changes Referee#3 commented "I am pleased with the edits and additions that Novi & colleagues have made to explain their approaches. I also agree with them that these approaches are useful for oceanography and biogeochemistry, and I hope myself to learn and implement them in my own work.".
We think, therefore, that the mathematical description is useful. We now explain the meaning of topology.

"I wonder if the title can more clearly reflect how the work uses stratification as a proxy to understand ocean oxygen and its evolution. Currently the manuscript sounds like a model intercomparison which it is not."
To address this new point we changed the title to:
"Evolution of oxygen, stratification and their relationship in the North Pacific Ocean in CMIP6 Earth System Models".

There are numerous errors in language use, generally in the use of gerunds and matching up singular/plural nouns and verbs. It makes parts of the manuscript quite difficult to understand. I cannot provide every example but I would strongly recommend the authors conduct a thorough proofreading for the manuscript.

We will proofread the manuscript again. It would have been helpful, however, to mention a few, as all authors have proofread the manuscript. We found few instances of docx to pdf conversion problems (which are not under our control) but not many typos.

**Introduction**

"There is no reference to current reconstruction schemes of ocean oxygen, or indeed the fact that observations of oxygen are sparse in space and time. As it is part of both the abstract and the discussion it should be included in the introduction."

In the previous review, the referee asked us to include a discussion on oxygen reconstructions in the discussion and conclusion part, which we acknowledged and included. Using the referee's words, they asked to restructure the discussion and conclusion section as follows: "I would recommend changing Section 5 to "Discussion and Conclusions" (or adding a separate Discussions section) and including a critical analysis on the relationship of this study to other work on the PDO and North Pacific oxygen (e.g., Ito et al 2019) or to reconstruction efforts (e.g. Sharp et al., 2022)." We addressed this concern (see our first answer report). To address this new point, we added a reference in the Introduction (Bindoff et al. 2019) together with a mentioning of the sparseness of oxygen observations and uncertainties associated with them.

"I appreciate the highlighting of the hypotheses and the questions in this new version, but the paper should be framed such that it references, in order:

1. The aim/objectives of the paper
2. The questions the paper aims to answer
3. The hypotheses for these questions

Right now the order is objective -> hypotheses -> questions, which makes it difficult to read. Additionally I do not think that the hypotheses need to be referenced as HYP1/2/3, as this just makes the manuscript more difficult to read and reference."

We re-arranged the sequence of our presentation.

We were asked in the previous review stage from this referee to make a clearer connection between the results and hypotheses and to add signposting to help the reader following this. To satisfy those comments, which we believe improved the manuscript presentation, we needed to add an explicit signpost to point the reader to the three hypotheses (HYP1/2/3), along with the major re-organization of the sections. We wish to keep them, as they were found useful by a different reviewer.

"There is no literature review on the use of IE in oceanography. Any citations about its application exist much later and they should absolutely be included in the introduction to provide justification for this work."

We now add references to IE in the Introduction.

"In addition, there needs to be a section in which the use of IE for physical properties like temperature is compared to its potential use for ocean biogeochemical properties. If space is an issue, this is something that could replace parts of the beginning discussion on ocean oxygen cycling which is a bit high-level, has very few references, and also mentions quite a few biological processes which are ignored for the rest of the manuscript."

We respectfully disagree with this comment (which we do not fully understand). Is the reviewer asking for a new section where we compare IE use for Temperature to its use for oxygen? Why? What is the motivation? This request was not pointed out as concern in the first review stage. What is the added value given the hypothesis to be tested in the manuscript?

"Lines 113-115 seem like an additional objective in the manuscript. If this is the case it needs to be reformatted."

We thank the referee for this comment. We moved the comment.

"Generally an introduction ends with an outline of the following sections. In the current introduction there is too much emphasis on the methods (which should be in the immediately-following section). A brief outline of what is included in the methods can be in the section introducing the following sections."

We respectfully notice that this is a personal preference and many papers in Biogeosciences do not include the outline at the end of the Introduction. We added a short outline, while not finding it useful.

**Materials and methods**

"Which historical experiments are being used? ESMs have emissions-based historical forcings and concentration-based historical forcings. In either case the radiative forcing is not necessarily the same between models with different atmospheric components and atmospheric chemistry."

This point is irrelevant to the main outcome of the manuscript, because the relationship between $O_2$ and stratification could display commonalities across models (different patterns potentially, but common behaviors) independently of the radiative forcing. However, we are more explicit in saying that we are using historical runs with $CO_2$ concentration-based historical forcing.

It would help to include which variables you take from the CMIP6 models. Also, please define how you calculate density from temperature and salinity and how it is implemented in Equation 4. Is it using EOS80 or TEOS10? Are all model outputs gridded so that 0-200m are full cells, or is there some interpolation involved?

TEOS10 (but in the upper 200 m differences will not really matter much). The models have different grids and levels. A linear interpolation is applied in the vertical and all datasets are bi-linearly interpolated to a 1x1 degree in the horizontal.

"I still find the names of these indices (Ind1, Ind2) to be very difficult to internalize, both in the methods section and afterwards. I would strongly suggest eliminating the shorthand entirely."

In the revised version, we already added a subscript to the indices to recall what changes they refer to: *means* for the changes in means, *variability* for the changes in varaibility, and *extremes* for the changes in exterems. We think that eliminating the shorthand entirely would burden the readability. We acowledge that it might not be the best format for all the readers, but the notation adopted was consistent with previous literature. We anyway replaced the names with $\Delta_{means}$, $\Delta_{variability}$ etc).

"For the definitions on lines 191-193 I would strongly suggest using an overbar to indicate time averages rather than angle brackets."

This is a matter of personal preference. We prefer angle brackets.

"I am still unsure about the arguments about the robustness of the extreme metric. If a 32 year time series of, say, winter stratification is used, the extreme is effectively the 97th percentile. This definition is necessarily dependent on both the length of the time series used and the number of ensemble members used. Additionally, is there a reason why you use seasonal averages to calculate the extremes, but then divide by the total number of months within that season? I am not sure if I am missing something major here but this continues to confuse me."

To compute the extremes we proceed as follows: (1) in each season of the first period, say 1950-1981, we compute the multi-year seasonal min (or max) using monthly data, not seasonal averages. This is the min (max) value ever reached by that variable in any of the three months considered for a given season. We set this value as threshold, then we count how many times (still at monthly frequency) over 1983-2014 the threshold is exceeded (above or below) in months belonging to that season. Therefore, as we work with monthly data, the following division is done over the total number of months considered. So that, for example, in winter we compare the number of times that a threshold is exceeded in the winter months over 1983-2014 (which we counted at monthly frequency), with the number of months contained in all the winters in that period, i.e. 3months*32 years.

We have verified that the definition is robust and different than using a fixed percentile. The reviewer may verify it as the codes are public. Indeed, the hotspot identification does not vary much with the percentile chosen (given the hotspot definition), or with the period selection, as shown below for the GFDL-ESM4 model.

[Figure]

Fig. 1 : Historical SED computed with different thresholds.

[Figure]

Fig. 2: Historical SED computed over different periods.

**Results**

The manuscript mentions that the hindcast "not surprisingly" displays the best fit for IPV and O2. This makes me wonder about the usefulness of a fully-coupled ESM compared to other ocean BGC models forced by reanalyses. What are the benefits of using a fully-coupled ESM? If the representation of the PDO is the largest issue in reproducing IPV and O2 (although I'm not convinced this is true) then I would assume this work would be more successful using ocean-only runs. Coupled models exhibit a "signal to noise paradox" (e.g., Scaife and Smith 2018, Zhang et al 2021) which results in an apparent decrease in predictability that may be restricted to model world. How would this impact the interpretability of your results?

Noting that none of the above issues were raised in the first review, we would like to point out that with the goals stated for this (our) manuscript, ocean only models would have not been useful. In addition:
1) We never say in the manuscript that the representation of the PDO is the largest issue. We say that even after removing the PDO signal (whichever it is) the O2 – stratification relationship changes across models and is not robust by and large outside the Equatorial region. We do say, however, that the O2 evolution has been shown to be influenced by the PDO, therefore the PDO focus.

2) E3SM-2G is introduced and used as reference model dataset (the best we can do with models run at climate-scale resolutions).

3) The benefit of using fully coupled ESMs is that they can project into the future, ocean only runs cannot, and we explicitly want to explore how the IPV-O2 relationship may evolve (and if such evolution displays common behaviors across models)

4) Coupled models (some at least) may exhibit a signal to noise paradox, but as it turns out their predictability potential varies greatly and indeed some models are more predictable (have overall lower entropy) than the reanalysis or hindcast counterpart (Figure 2).

Considering the detailed analysis presented in Falasca and Bracco (2022) on two models, coupled climate models continue to be more predictable than observations whenever predictability is defined on their topological properties. The papers cited by the reviewer apply different definitions of predictability to specific processes that differ from what we are exploring here.

I am a bit confused about the argument about how low-frequency PDO modulation alongside higher-frequency variability can produce low predictability. By that argument there is information associated with the PDO, but it is just more difficult to isolate. Is this not a case where the timescales of interest need to be controlled for? You use interannual changes in seasonal values for IPV and oxygen, but if we know the PDO to operate on decadal timescales, is there any way that the IE method can be altered to take these longer timescales into account?

The method is looking for recurrences. At those scales, a signal that emerges from the superposition of different time scales (noise at short time scales and a quasi-sinusoidal modulation at long time scales) will show very little predictability at intermediate time scales – those of interest here. It is possible in principle to apply a running mean - let's say decadal- on each grid point of the domain and each variable, and re-run the IE analysis to isolate the PDO signal, but this strategy will require much longer time-series (a couple of centuries at least) to be meaningful and will not answer the questions posed in this manuscript. We filtered/isolated the PDO signal instead.

"Please ensure that all figures follow inclusivity guidelines. In particular Figures 4, 5, 9, and 13 need to be revised so that readers with deuteranomaly (red-green colourblindness) can interpret the figures. "

Thanks for the comment. All figures were tested already with the free software *Coblis – Color Blindness Simulator* to verify that they are visible by deuteranomaly, protanomaly, tritanomaly and blue cone monocromacy. We report below how the software shows the figures. We only show variations for Fig. 13 and for Fig. 4 because Fig. 5 has the same color palette as Fig. 4, and Fig. 9 has the same colorpalette of Fig. 13.

[Figure]

Fig.3: variations of Fig 13 (same colorpalette as Fig. 9 of the submitted manuscript)

[Figure]

Fig. 4: Variations of Fig.4 (same color palette of Fig. 5 of the Submitted manuscript)

[Figure]

Fig.5: variation of Fig.4 (Same color palette as Fig.5 of the submitted manuscript)

Additional points on figure legibility:

● Figure 1: Please indicate which panels illustrate biases within the figure itself and ensure that "RMSE = " is included in each subplot.

We modified the figure accordingly (the title lines got too busy though, which required introducing shorter names for the models)

● Figure 2: The years for the historical and future runs should be included in the figure, as well as some reference to IE so that the reader can differentiate the figure from later figures. Also, the colorbar direction seems counterintuitive to me, as I would normally associate green with "good" (i.e., lots of information) and this is actually for high entropy.

The reason why the years are not explicitly included in the figure(s) is to avoid redundancy. The "historical" period– as stated multiple times in the manuscript and in the caption – is 1950-2014 for the modes but 1960-2014 for the hindcast and the reanalysis. Adding the information in the figure would force us to either (i) repeat the period atop of each panel or (ii) adding the period below the title but specifying that it is 1960-2014 for hindcast and reanalysis. In both cases the figure would be contain a lot of text without adding any information as it is repeatedly stated in caption and text.
The colorbar direction is a personal preference. Also, the colorbar has not changed since the first submitted version and was not problematic in the previous version.
The reference to IE is given in the caption but we added a title to explicitly indicate it.

● Figure 3: Change the x-ticks for the historical run to green and the future scenario run to black. Subpanel b needs a similar label to differentiate the green and black dashed lines.
Thank you for this comment, we modified as requested.

● Figures 4 & 5: the contours for ENSO and PDO are difficult to separate for some panels. Please use dashed lines for one of the indices so that the reader can more easily discern which

regions are related to ENSO and PDO.
Thank you for this comment, we modified as requested.

● Figure 6: The years in the caption are wrong and they should reflect the two periods used in the averaging rather than combine them into one period.
We modified the caption but we retain the period as it is correct (while divided in 2 for the calculation as explained in the text).

**Discussion/Conclusions**

Thanks for including a discussion. I think there still needs to be many more citations. For instance, the first page cites only Ito et al 2019. On line 534-535 you reference a wide range of mechanisms but there is no relevant literature included.

We repeated some of the references contained in the Introduction and added some new ones.

In the discussion, the manuscript references that reconstructions need to interpolate onto a regular grid. This is fleshed out more in the authors' response. This is not true – there is no inherent requirement to interpolate observations *or* innovations onto a regular grid when creating a reconstruction. There is utility in doing this, particularly as reconstructions are often used to compare or benchmark models, but any further assertion should be removed from the manuscript.

We thank the referee for noticing this, we will reframe that sentence to avoid any superflous implication. However, the sentence is -we believe – for the IE calculation (not reconstructions), which can be performed on ARGO data interpolated to a regular grid and cannot be performed as defined on quasi-Lagrangian observations.

Please include a concluding paragraph that restates the objectives and main findings of the paper.
Thank you for the recommendation, we added what requested.